# A first year-long estimate of the Paris region fossil fuel $CO_2$ emissions based on atmospheric inversion

Johannes Staufer[1], Grégoire Broquet[1], François-Marie Bréon[1], Vincent Puygrenier[1], Frédéric Chevallier[1], Irène Xueref-Rémy[1], Elsa Dieudonné[1,2], Morgan Lopez[1,3], Martina Schmidt[1,4], Michel Ramonet[1], Olivier Perrussel[5], Christine Lac[6], Lin Wu[1], and Philippe Ciais[1]

[1]Laboratoire des Sciences du Climat et de l'Environnement, LSCE/IPSL, CEA-CNRS-UVSQ, Université Paris-Saclay, Gif-sur-Yvette, France
[2]now at Laboratoire de Physico-Chimie de l'Atmosphère, Université du Littoral, Côte d'Opale, Dunkerque, France
[3]now at Environment and Climate Change Canada, Toronto, Canada
[4]now at Institut für Umweltphysik, Ruprecht-Karls-Universität Heidelberg, Heidelberg, Germany
[5]AIRPARIF Surveillance de la Qualité de l'Air en Île-de-France, Paris, France
[6]CNRM-GAME (CNRS-Météo-France), UMR3589, Toulouse, France

*Correspondence to:* J.Staufer (johannes.staufer@lsce.ipsl.fr)

**Abstract.** The ability of a Bayesian atmospheric inversion to quantify the Paris region's fossil fuel $CO_2$ emissions on a monthly basis, based on a network of three surface stations operated during one year as part of the CO2-MEGAPARIS experiment (August 2010–July 2011), is analysed. Differences in hourly $CO_2$ atmospheric mole fraction between the near-ground monitoring sites ($CO_2$ gradients), located at the north-eastern and south-western edges of the urban area, are used to estimate the 6- h mean fossil fuel $CO_2$ emission. The inversion relies on the CHIMERE transport model run at $2\,\text{km} \times 2\,\text{km}$ horizontal resolution, on the spatial distribution of fossil fuel $CO_2$ emissions in 2008 from a local inventory established at $1\,\text{km} \times 1\,\text{km}$ horizontal resolution by the AIRPARIF air quality agency, and on the spatial distribution of the biogenic $CO_2$ fluxes from the C-TESSEL land surface model. It corrects a prior estimate of the 6- h mean budgets of the fossil fuel $CO_2$ emissions given by the AIRPARIF 2008 inventory. We found that a stringent selection of $CO_2$ gradients is necessary for reliable inversion results, due to large modelling uncertainties. In particular, the most robust data selection analysed in this study uses only mid-afternoon gradients if wind speeds are larger than $3\,\text{ms}^{-1}$ and if the modelled wind at the upwind site is within $\pm 15^o$ of the transect between downwind and upwind site. This stringent data selection removes 92 % of the hourly observations. Even though this leaves few remaining data to constrain the emissions, the inversion system diagnoses that their assimilation significantly reduces the uncertainty in monthly emissions, by 9 % in November 2010 to 50 % in October 2010. The inverted monthly mean emissions correlate well with independent monthly mean air temperature. Furthermore, the inverted annual mean emission is consistent with the independent revision of the AIRPARIF inventory for the year 2010, which better corresponds to the measurement period than the 2008 inventory. Several tests of the inversion's sensitivity to prior emission estimates, to the assumed spatial distribution of the emissions, and to the atmospheric transport modelling demonstrate the robustness of the measurement constraint on inverted fossil fuel $CO_2$ emissions. The results, however, show significant sensitivity to the description of the emissions' spatial distribution in the inversion system, demonstrating the need to rely on high-resolution local inventories such as that from AIRPARIF. Although the inversion constrains emissions through the assimilation of $CO_2$ gradients, the results are

hampered by the improperly-modelled influence of remote $CO_2$ fluxes when air masses originate from urbanised and industrialised areas north-east of Paris. The drastic data selection used in this study limits the ability to continuously monitor Paris fossil fuel $CO_2$ emissions: the inversion results for specific months such as September 2010 or November 2010 are poorly constrained by too few $CO_2$ measurements. The high sensitivity of the inverted emissions to the prior emissions' diurnal variations highlights the limitations induced by assimilating data during afternoon only. Furthermore, even though the inversion improves the seasonal variation and the annual budget of the city's emissions, the assimilation of data during a limited number of suitable days does not necessarily yield robust estimates for individual months. These limitations could be overcome through a refinement of the data processing for a wider data selection, and through the expansion of the observation network.

# 1 Introduction

There is a high political and scientific interest in developing methods for improving and verifying estimates of fossil fuel and cement $CO_2$ emissions. Consequently, there is an increasing deployment of urban $CO_2$ monitoring networks with the objective of quantifying city emissions through the atmospheric inversion approach (Boon et al., 2016; Duren and Miller, 2012; Lauvaux et al., 2013, 2016; Kort et al., 2013; McKain et al., 2012; Strong et al., 2011; Turnbull et al., 2015). Bréon et al. (2015), upon which this study builds, recently reported first estimates of fossil fuel $CO_2$ emissions of the Paris urban area during a two-month period. They used three ground-based $CO_2$ measurement sites at the north-eastern and south-western edge of the area and an inversion system based on a $2\,km \times 2\,km$ horizontal resolution transport model. The monitoring stations in Gonesse (GON), approximately $15\,km$ north of Paris' city centre, and in Montgé-en-Goële (MON), $35\,km$ north-east (NE) of Paris' city centre, were deployed by the CO2-MEGAPARIS project and operated from August 2010 to July 2011. The monitoring station in Gif-sur-Yvette (GIF), $20\,km$ south-west (SW) of Paris' city centre, is part of the Integrated Carbon Observation System-France long-term network.

The main principle of the atmospheric inversion proposed by Bréon et al. (2015) consists in constraining $CO_2$ emission budgets of the urban area by assimilating atmospheric $CO_2$ mole fraction gradients between pairs of sites located upwind and downwind of the city. The use of cross-city gradients, rather than individual mole fractions, aims at eliminating the variability of $CO_2$ caused by the transport of remote and natural fluxes outside the urban area. It assumes that the signal from these fluxes has a relatively large spatial and temporal scale compared to the distance and transport duration between the measurement sites. These signals and the potential signal from natural fluxes within the urban area cannot be sufficiently well controlled by the monitoring network, in particular because their large day-to-day variations cannot be filtered as a smooth baseline in the time series of $CO_2$ concentrations at individual sites (an approach frequently used in regional atmospheric inversions, e.g., in Henne et al., 2016). On the contrary, such signals can be as high as the signal caused by the emissions within the urban area (Bréon et al., 2015; Kort et al., 2013; Nordbo et al., 2012). Uncertainties in remote and natural fluxes can thus highly impact the skill for inverting the urban emissions. In the simulations by Bréon et al. (2015), the ratio between the signal from the natural and remote fluxes and the signal from the urban emissions is high when analysing individual measurements. It, however, strongly decreases when analysing gradients. This weak impact of natural fluxes on inversions is on one hand due to

the fact that the dense and compact Paris urban area exhibits little vegetation within its bounds. On the other hand, it is due to the fact that, upwind the city, the signal from fluxes outside this urban area is sufficiently diffused in space so that it is relatively homogeneous over the Paris urban area and constant during the duration of the transport over this area.

The selected cross-city gradients also provide a characterization of the increase in the $CO_2$ mixing ratios of air parcels that pass over the city. It is assumed that these gradients represent emissions from the entire city and are not highly sensitive to the distribution of the emissions. This assumption is line with the inversion system of Bréon et al. (2015) that controls the city-scale emissions budgets and the temporal variation of fossil fuel $CO_2$ emissions, but not their spatial distribution. However, this method should not be seen as a sort of mass balance, given that, in practice, the inversion is not set up to ensure that the upwind and downwind concentrations corresponds to the same air masses that travelled from the upwind to the downwind site. Furthermore, since the atmospheric boundary layer evolves significantly in space and time and due to the atmospheric diffusion during the transport over the Paris area, such cross-city gradients cannot perfectly represent the $CO_2$ enrichment of air parcels passing over the urban area. In addition, temporal variations of the emissions during the transport of air masses over the Paris area prevent from relating a given gradient to the emissions at a given time. The gradients need to be interpreted using a transport model and knowledge on the spatio-temporal variations of the emissions at hourly scale. In that sense, the assimilation of gradients is affected by transport modelling uncertainties and by uncertainties in the variations of the emissions at high spatial and temporal resolution, such as it the case for any inverse modelling approaches.

The inversion assimilates cross-city $CO_2$ gradients during afternoon to correct prior estimates of 6-h fossil fuel $CO_2$ emissions budgets of the Paris metropolitan area (Île-de-France administrative region). These prior estimates are derived from the AIRPARIF inventory for the year 2008 (AIRPARIF, 2012). AIRPARIF is a non-profit agency that is accredited by the French Ministry of Environment to monitor the air quality in Île-de-France. Even though they have a limited impact on the inversion when gradients are assimilated, the system of Bréon et al. (2015) also inverts biogenic fluxes and corrects prior estimates of the biogenic fluxes from C-TESSEL, the land-surface component of the ECMWF (European Centre for Medium-Range Weather Forecasts) numerical weather forecasting system (Boussetta et al., 2013). In order to model the $CO_2$ gradients, the inversion uses an estimate of the fossil fuel $CO_2$ emission and biogenic flux distribution at $2\,km\times2\,km$ and hourly resolution, coupled to a $2\,km\times2\,km$ resolution configuration of the chemistry transport model CHIMERE (Menut et al., 2013).

Bréon et al. (2015) developed and tested this inversion set-up for two months in autumn 2010. The values of the AIRPARIF 2008 inventory were used to derive the prior estimates for the corresponding dates in 2010. Bréon et al. (2015) reported a significant improvement of the fit between modelled and measured $CO_2$ gradients by the inversion and reasonable patterns of corrections applied to prior emission values. The small number of monitoring sites and the stringent criteria for selecting gradients leads to a high number of periods, ranging from one to several days, during which the inversion does not assimilate any atmospheric $CO_2$ data. As a consequence, averages of the inverted emissions over one month were found to be more reliable than 1-day to 1-week mean results.

The aim of this study is to derive a full year of monthly mean emission estimates for the Paris area, based on the inversion system described by Bréon et al. (2015) and on the availability of measurements at MON, GON and GIF during the period mid-2010 to mid-2011. The 1-yr long inversion allows a better evaluation of the method by analysing the seasonal variation

and the annual budget of the inverted emissions. In particular, the annual budget can be compared to the AIRPARIF emission assessment for 2010 (AIRPARIF, 2013). This assessment is based on an inventory model that has been improved since the release of the 2008 inventory. The 2010 inventory applies to a time period which better corresponds to the inversion period than the 2008 inventory used for the inversion. Therefore it provides some independent information to check the corrections applied by the inversion to the prior estimate of the annual budget derived from the 2008 inventory.

Preliminary tests of the inversion during the 1-yr period, however, revealed that the selection of gradients, as proposed by Bréon et al. (2015), do not conform fully with the underlying assumptions of having gradients dominantly influenced by urban emissions. Notably, negative $CO_2$ gradients between downwind and upwind sites were frequently measured when using the gradient selection criteria of Bréon et al. (2015). This led us to revise the selection of $CO_2$ data to form gradients. The revision consists primarily on a tighter filtering of wind directions to select gradients in order to avoid situations when air parcels leaving the upwind site or reaching the downwind site do not overpass a significant part of the city and the vicinity of the other site. Section 2 presents a summary description of the inversion configuration and the revised gradient selection. Section 3 analyses the inversion results for different configurations. In particular, it assesses the impact of the stricter gradient selection, and the sensitivity of the results to the prior emission estimates, to the emissions' spatial distribution, and to the atmospheric transport modelling, so as to evaluate how robustly the emissions are constrained by atmospheric $CO_2$ data. These results are discussed in Sect. 4.

## 2   Inversion configuration

The inversion method described by Bréon et al. (2015) is based on the Bayesian approach. The control vector $x$ gathers the $CO_2$ flux budgets. $x_b$ is the vector of the prior estimates of these budgets, independent of atmospheric observations. The observed $CO_2$ mole fraction gradients selected for the inversion are assembled into $y_0$, which defines the observation space $y$. The linear observation operator $\mathcal{H} : x \mapsto y = \mathbf{H}x + y^{\mathrm{f}}$ projects the control vector $x$ into the observation space $y$ through the linear operator $\mathbf{H}$ (combining the description of the fluxes' spatial distribution and the atmospheric transport model) and the addition of $CO_2$ gradients $y^{\mathrm{f}}$ caused by fluxes that are not controlled by the inversion, such as the remote fluxes characterized by the $CO_2$ boundary conditions of the regional transport model. The uncertainties in $x_b$ and the observation errors, i.e., errors in the measurements $y_0$ and from the observation operator $\mathcal{H}$, are assumed to have unbiased Gaussian distributions and are characterized by the prior uncertainty covariance matrix $\mathbf{B}$ and the observation error covariance matrix $\mathbf{R}$, respectively. $x_a$, the optimal posterior estimate of $x$, knowing its prior estimate $x_b$ and measurements $y_0$, can be obtained from (e.g., Rodgers, 2000):

$$x_a = x_b + (\mathbf{B}^{-1} + \mathbf{H}^{\mathrm{T}}\mathbf{R}^{-1}\mathbf{H})^{-1}\mathbf{H}^{\mathrm{T}}\mathbf{R}^{-1}(y_0 - y^{\mathrm{f}} - \mathbf{H}x_b). \tag{1}$$

The uncertainty in $x_a$ has unbiased Gaussian distribution and is characterized by the posterior uncertainty covariance matrix $\mathbf{A}$:

$$\mathbf{A} = (\mathbf{B}^{-1} + \mathbf{H}^{\mathrm{T}}\mathbf{R}^{-1}\mathbf{H})^{-1}. \tag{2}$$

The inversion uses measurements during a given 30-day period to derive fluxes during the same 30-day period. Independent
inversions are made for twelve consecutive 30-days periods starting on 1 August 2010 to cover the entire observation period
from August 2010 to July 2011. The 6-h mean inverted emissions during each period serve as the basis for the analysis of
emissions in the Paris area at the monthly scale. Even though these 30 day periods do not correspond exactly to the calendar
months, the names of the calendar months are used to label them.

We briefly recall descriptions of the components of Eq. (1)–(2) as laid out by Bréon et al. (2015) in the next Sections
(Sect. 2.1-2.4). As detailed in Sect. 2.2, two modifications, however, are brought to the definition of the observation space $y$
and thus to the observation operator $\mathcal{H}$. The modifications result in two new inversion configurations that are denominated
*initial* (i.e., close to Bréon et al., 2015) and *reference* configuration hereafter. Section 2.6 presents the set-up of the sensitivity
tests, where the prior estimates of the control variable, $x_b$, and components of the observation operator $\mathcal{H}$ are modified with
respect to the reference configuration.

## 2.1 Control vector $x$ and the prior estimate of the flux budgets $x_b$

$x$ contains 6-h mean fossil fuel $CO_2$ emission budgets for windows 0-6 h, 6-12 h, 12-18 h, 18-24 h (local time is used hereafter)
for each day for the Île-de-France region. Most of the emissions in this region are concentrated in the urban agglomeration of
Paris. Thus, this choice of $x$ approximately consists in controlling the emission budget of this urban area. $x$ also contains 30-
day mean biogenic $CO_2$ fluxes for each of the four 6-hour windows of the day (0-6 h, 6-12 h, 12-18 h, 18-24 h) for 9 areas that
make up the Northern France modelling domain, including one that encompasses the Paris region (see Fig. 1). The inversion
optimises the diurnal cycle of both the fossil fuel $CO_2$ emissions and biogenic fluxes through resolving these fluxes for the
different 6-h windows of the day. However, it controls the day-to-day variability of the fossil fuel $CO_2$ emissions but not the
one of the biogenic fluxes. The inversion controls scaling factors of the flux budgets provided by the emission inventories and
the ecosystem model simulations through the linear part of the observation operator ($\mathbf{H}$, see below). For the sake of simplicity
we state hereafter that it controls the flux budgets themselves.

The initial and reference inversion configurations use our best available knowledge on the flux budgets—the AIRPARIF 2008
inventory (since the AIRPARIF 2010 monthly mean budgets were not available for this study) and the C-TESSEL simulation—
to define the prior estimate $x_b$. The sensitivity tests, described in Sect. 2.6, investigate the impact of using different prior
estimates for the Paris fossil fuel $CO_2$ emissions.

## 2.2 Configuration of the observation vector $y$ and measurement vector $y_0$

The specific definition of the observation space $y$ and of the corresponding measurement vector $y_0$ depends on the measurement availability, on the range of wind directions used to select gradients, and on the meteorological forcing of the $CO_2$ transport model. Two different meteorological products are used to define the wind direction for the gradient selection (see forward Sect. 2.3.1).

The three monitoring sites are located roughly along a NE-SW direction at edges of the urban area in mixed urban-rural environments (Fig. 1) at heights of $9\,m$ (MON), $4\,m$ (GON) and $7\,m$ (GIF) above ground. The NE-SW direction corresponds to the dominant wind directions in Île-de-France. Technical details about the measurements are given in Xueref-Remy et al. (2016), Bréon et al. (2015) and Lac et al. (2013). Here we briefly summarise the main aspects. The CO2-MEGAPARIS sites GON and MON were equipped with a ring-down cavity analyser from Picarro (model G1302), while an automated gas chromatograph analyser (Agilent HP6890, see Gibert et al., 2007) has been used at GIF. All measurements are quality controlled and calibrated against the World Meteorological Organisation mole fraction scale WMO-X2007 (Zhao and Tans, 2006). The instrumental reproducibility of the Piccaro $5\,min$ averages is better than $0.17\,ppm$, while measurement accuracy is estimated at $0.38\,ppm$ (Bréon et al., 2015; Xueref-Remy et al., 2016). The precision of the chromatograph analyser in GIF is estimated at $0.05\,ppm$ for $5\,min$ averages (Lac et al., 2013). In our study, we binned measured $CO_2$ data into 1-h means. The accuracy for these hourly means is better than $0.4\,ppm$ at the three sites which is negligible compared to the modelling uncertainties (see forward Sect. 2.5).

Figure 2 and Fig. A1–A2 illustrate the temporal coverage of the measurements available during the CO2-MEGAPARIS period (August 2010–July 2011) at each measurement site. They also show which data are finally used to form gradients. Some significant data gaps can be noticed, e.g., during June 2010 and 2011 at GON, September 2010 at MON, January, November and December 2010 at GIF. The regular 1-day gaps correspond to instrument calibrations.

$CO_2$ at the measurement sites is significantly influenced by both the Paris urban emissions and the remote fluxes (i.e., by fluxes outside the modelling domain, whose influence is simulated by the transport of the $CO_2$ conditions imposed at the model boundaries, and by biogenic and fossil fuel fluxes within the modelling domain but outside the Paris urban area). It is assumed, that, due to atmospheric diffusion, the signature of the remote fluxes upwind the city on the concentrations in our domain has horizontal and vertical spatial scales and a temporal scale of variability that are large enough so that it does not evolve during the transit of an air parcel above the city. In other words, it is assumed that the remote fluxes do not cause $CO_2$ gradients between downwind and upwind stations when the wind blows from the upwind to the downwind sites. This critical assumption is supported by the fact that the simulated $CO_2$ gradients, caused by remote fluxes, are negligible. However, this does not necessarily imply that the measured gradients are not influenced by the actual fluxes (Bréon et al., 2015). This assumption is also supported by the much better fit between observed and modelled $CO_2$, when observations are defined by cross-city gradients instead of $CO_2$ mixing ratios at individual sites (Bréon et al., 2015). By assimilating $CO_2$ gradients rather than individual $CO_2$ mole fractions, we thus expect to prevent the inversion from being sensitive to the uncertainties in the estimate of the remote fluxes.

Local sources in the vicinity of the measurement sites are difficult to represent in the model. In order to limit their impact, Bréon et al. (2015) selected gradients only if the wind speed is above a given threshold of $2\,\mathrm{m\,s^{-1}}$. Similar to most inversion studies that used rural measurement sites (e.g., Broquet et al., 2011; Geels et al., 2007), Bréon et al. (2015) assimilated data during the afternoon only, since the model seemed to poorly represent vertical transport during other periods of the day.

Specifically, Bréon et al. (2015) used differences in simultaneous hourly-averaged $CO_2$ measurements between the peri-urban stations during the afternoon (12-16 h) to define the measurement vector $\boldsymbol{y}_0$. When (at a given hour) the wind at GIF, given by the meteorological simulation (see below Sect. 2.3.1), is from SW, i.e. from $160^o$ to $260^o$, and is above $2\,\mathrm{m\,s^{-1}}$, GIF is the upwind site and Bréon et al. (2015) assimilate hourly $CO_2$ mole fraction differences between MON and GIF and between GON and GIF. When the simulated wind at MON is from NE, i.e. from $0^o$ to $135^o$ and exceeds $2\,\mathrm{m\,s^{-1}}$, MON is the upwind

site and Bréon et al. (2015) assimilate the $CO_2$ differences between GON and MON and between GON and GIF.

Using this configuration, Bréon et al. (2015) assimilated $CO_2$ gradients between GON and MON. GON and MON are separated by only a short distance. The enhancement of $CO_2$ between these two sites therefore rather reflects the emissions from a small portion of the North-Eastern suburbs of Paris than emissions from the entire urban area. Model-data misfits for such gradients relate far more to the uncertainties in the high resolution mapping of the emissions than to uncertainties in

the budget of the city emissions. In addition, these gradients are strongly affected by emissions from the Charles-de-Gaulle airport which is located between the two sites and an important local source of $CO_2$ that is not representative of the main $CO_2$ sources in the Paris urban area. Thus, gradients between GON and MON are not adapted to constrain city-scale emissions. Furthermore, in order to retain a significant fraction of measurements in the inversion, Bréon et al. (2015) used a loose range of wind directions to define upwind and downwind conditions. This loose range could allow the assimilation of gradients when air

masses leaving the upwind site or reaching the downwind site hardly cross a significant portion of the Paris urban area, or, more generally, when air masses are not really transported from the upwind to the downwind site. This loose selection of gradients for constraining fluxes was not identified as a major source of systematic error. Through this configuration, Bréon et al. (2015) primarily aimed at decreasing the impact of remote fluxes on $CO_2$ mole fractions while keeping a large amount of data for the inversion. Both choices, the assimilation of GON-MON and the loose wind filtering to select gradients , however, lead to

estimates of spatially integrated emissions of the city constrained by measurements that are influenced only by emissions from a small fraction of the city. This would not be an issue if the spatial distribution of emissions provided by AIRPARIF was perfectly accurate. On the other hand, any significant error in the emissions' spatial distribution may induce a large error on the city-wide emission inversion. Indeed, if the assumed spatial distribution of the emission bears significant errors (which is likely the case), the inversion corrections, driven by model-data misfits due to errors in emissions from a small part of the city,

will become inconsistent with the errors at the city-scale, raising large so-called aggregation errors (Kaminski et al., 2001).

As mentioned in the introduction, preliminary tests of inversions using the configuration of Bréon et al. (2015) for the period August 2010–July 2011 demonstrated the need for an improved configuration where the selection of $CO_2$ conforms better with our assumptions on gradients. In this study, two critical changes are applied. They consist in assimilating GIF-GON and GIF-MON and in discarding GON-MON gradients when the wind is from NE. Furthermore, a stricter (narrower) range of wind

directions to select $CO_2$ gradients is used. Discarding GON-MON gradients suppresses the large amount of negative gradients

in the measurement vector $\boldsymbol{y}_0$. The impact of discarding GON-MON gradients on the inversion results is not analysed deeper in the following. It relates to specific details of the Paris network configuration. Here, we focuses on the impact of using a narrower range of wind directions for the gradient selection. The stricter selection of wind directions consists in assimilating a gradient between two sites only if the modelled wind at the upwind site is within $\pm15^o$ of the transect between the downwind and upwind site. The specific choice of $\pm15^o$ is somewhat arbitrary. On the one hand it ensures the selection of a significant number of gradients. On the other hand it ensures that air masses leaving the upwind site or reaching the downwind site are transported over a large part of the urban area and in a direction that is close to the transect between downwind and upwind sites. Thus, the gradients GIF-GON, GIF-MON, MON-GIF and GON-GIF are assimilated only if the wind is from $20^o$ to $50^o$, $35^o$ to $65^o$, $215^o$ to $245^o$, and $200^o$ to $230^o$, respectively. We use the term *SW gradients* for the gradients GON-GIF and MON-GIF and *NE gradients* for the gradients GIF-MON and GIF-GON.

We apply other significant changes to the gradient selection criteria of Bréon et al. (2015). First, we increase here the minimum wind speed threshold at the upwind site from 2 to $3\,\mathrm{ms}^{-1}$. This change is driven by the fact that, as noticed by Bréon et al. (2015), large model-data misfits persist after inversion for wind speeds close to $2\,\mathrm{ms}^{-1}$. This suggests that a threshold of $2\,\mathrm{ms}^{-1}$ was not sufficient to avoid a large contamination of the measurements by poorly-modelled local sources. Furthermore, in this study, a single valid 1- h mean gradient during a given afternoon is not selected for the inversion. This avoids constraining the emissions of a given day based on a single observation that potentially bears a large transport model error. At last, an analysis of the impact of individual observations on the corrections applied by the inversion to the prior monthly flux estimates (i.e., impact of the product between the gain matrix $\mathbf{K} = (\mathbf{B}^{-1} + \mathbf{H}^{\mathrm{T}}\mathbf{R}^{-1}\mathbf{H})^{-1}\mathbf{H}^{\mathrm{T}}\mathbf{R}^{-1}$ and the model-data misfit for each individual gradient, see for more details Moore et al., 2011) was conducted for the initial and reference inversion experiments. It revealed that, for the initial inversion, during November, two gradients had far more impact on the correction to the emissions budget of this month than the other gradients. The gradients removed had both an impact of approximately -0.3 $\mathrm{MtCO_2}$ on this budget (i.e. approximately -0.6 $\mathrm{MtCO_2}$ in total). In both cases, these high impacts were connected to high prior model-data misfits during weak vertical mixing episodes. Again, similar to many inversion experiments (e.g, Chevallier et al., 2010), in order to avoid giving too much weight to individual measurements the two corresponding gradients were removed from the initial inversion experiment. However, such gradients are not selected by the tighter wind direction filtering of the reference inversion.

Three configurations of the observation space $\boldsymbol{y}$ are used in this study: $\boldsymbol{y}_{ini}$, $\boldsymbol{y}_{ref}$, and $\boldsymbol{y}_{lag}$. The first one, $\boldsymbol{y}_{ini}$, corresponds to the initial inversion configuration. It includes all the new options discussed above, except the narrowing of the wind direction ranges for the gradient selection. The selection of GIF-GON, GIF-MON, MON-GIF and GON-GIF gradients in $\boldsymbol{y}_{ini}$ is based on the wind direction ranges at GIF and MON as proposed by Bréon et al. (2015). The second one, $\boldsymbol{y}_{ref}$, corresponds to the reference inversion configuration. It includes all the new options and selects gradients based on the new wind direction ranges at GIF, GON and MON defined in this section. The comparison between the initial and reference inversions is used to assess the impact of using tight wind direction ranges on retrieved emissions and to evaluate if the selected gradients now conform better with our assumptions (see Sect. 3.1-3.2). $\boldsymbol{y}_{lag}$ is only used for a single experiment whose results are briefly discussed in Sect. 3.2. This observation vector consists in spatio-temporal gradients, i.e., mole fraction differences between a downwind

site at a given time and an upwind site 2 h before. Given a mean wind speed of $7\,\mathrm{ms^{-1}}$ in the lower planetary boundary layer of the Paris area during the afternoon over the 1-yr period and a distance of about $40\,\mathrm{km}$ between the upwind and downwind site the typical time for air being transported from the upwind to the downside site is approximately 2 h. The wind selection in this experiment is similar to that of the reference experiment. It uses simulated wind fields at the time of the upwind mole fraction

measurement involved in the gradient. At a given site the assimilation window is also reduced so that a given gradient does not involve any measurement outside the 12-16 h window, despite the use of a time lag in the gradient. The use of spatio-temporal gradients instead of spatial gradients appears more in line with the concept of a mass balance approach which constrains emissions based on mole fraction variations in air parcels that are transported over the Paris area. Due to atmospheric diffusion and variations in the planetary boundary layer, the spatio-temporal gradients still need to be interpreted using a high resolution

transport model. However, with such a configuration of the observation vector, the number of data that can be assimilated is further decreased as the assimilation window at both the upwind and downwind sites is reduced.

## 2.3   Observation operator $\mathcal{H}$

This section describes the observation operator $\mathcal{H} : \boldsymbol{x} \mapsto \boldsymbol{y} = \mathbf{H}\boldsymbol{x} + \boldsymbol{y}^{\mathrm{f}}$. The linear operator $\mathbf{H}$ can be decomposed into three operators ($\mathbf{H} = \mathbf{H}^{\mathrm{samp}}\mathbf{H}^{\mathrm{trans}}\mathbf{H}^{\mathrm{map}}$) consisting in the fluxes' spatio-temporal distribution ($\mathbf{H}^{\mathrm{map}}$), the atmospheric transport sim-

ulated using CHIMERE ($\mathbf{H}^{\mathrm{trans}}$), and the sampling of simulated 4D-$CO_2$ field like the observations ($\mathbf{H}^{\mathrm{samp}}$). $\boldsymbol{y}^{\mathrm{f}}$ gathers influences on the gradients which are not controlled by the inversion such as the signature of the model boundary conditions. In the following, we present the implementation of these operators and vectors used in Bréon et al. (2015) and the initial and reference inversions of this study, respectively, as well as alternative options used for sensitivity tests.

### 2.3.1   Atmospheric transport modelling and sampling

The atmospheric transport $\mathbf{H}^{\mathrm{trans}}$ and the signature of sources and sinks on $CO_2$ concentrations that are not controlled by the inversion ($\boldsymbol{y}^{\mathrm{f}}$) are modelled using a Northern France configuration of CHIMERE. It has a a $2\,\mathrm{km}{\times}2\,\mathrm{km}$ spatial resolution for the Paris region, and a $2{\times}10\,\mathrm{km}$ and $10\,\mathrm{km}{\times}10\,\mathrm{km}$ spatial resolution for the surroundings (see Fig. 1). It has 20 vertical hybrid pressure-sigma (terrain-following) layers that range between surface and the mid-troposphere, up to $500\,\mathrm{hPa}$. In the initial and reference inversion of this study, as in Bréon et al. (2015), CHIMERE is driven by operational analyses of ECMWF's Integrated

Forecasting System, available at approximately $15\,\mathrm{km}{\times}15\,\mathrm{km}$ spatial resolution and 3 h temporal resolution. In this case we will denote $\mathbf{H}^{\mathrm{trans}}{=}\mathbf{H}^{\mathrm{trans}}_{ECM}$ and $\boldsymbol{y}^{\mathrm{f}} = \boldsymbol{y}^{\mathrm{f}}_{ini-ECM}, \boldsymbol{y}^{\mathrm{f}}_{ref-ECM}$ or $\boldsymbol{y}^{\mathrm{f}}_{lag-ECM}$, depending on the type of gradient selection used.

Lac et al. (2013) conducted meteorological simulations on our modelling domain using a $2\,\mathrm{km}{\times}2\,\mathrm{km}$ resolution configuration of the non-hydrostatic mesoscale model Meso-NH. Meso-NH, jointly developed by Météo-France and Laboratoire d'Aérologie (Lafore et al., 1998), is coupled to 3-hourly analysed meteorological fields from AROME (Application of Re-

search to Operations at Mesoscale)-France (Seity et al., 2010) and to the land-surface-atmosphere interaction model SURFEX (Masson et al., 2013). SURFEX includes the urban and vegetation scheme TEB (Masson, 2000). Therefore, in contrast to the ECMWF meteorological forcing, Meso-NH/TEB includes some urban parametrisation, which may have a large impact on the transport over the city. Lac et al. (2013) showed, by comparison to Lidar systems operated on a short-term basis in the Paris

area, that Meso-NH/TEB captures relatively well the diurnal cycle of the boundary layer height as well as the differences in this height between peri-urban and urban locations.

A test of sensitivity is conducted to assess the impact of the uncertainties in the meteorological product on the inversions (in particular the uncertainties in the wind and in the boundary layer height). The meteorological product is used to drive the atmospheric transport model and to select the cross-city gradients. This test consists in using hourly mean outputs of Meso-NH/TEB to drive CHIMERE and to select gradients. Meso-NH/TEB simulations, originally conducted over a slightly different grid (see Lac et al., 2013, their Fig. 1a.), are interpolated onto the CHIMERE grid. When using Meso-NH/TEB $\mathbf{H}^{\text{trans}}$ and $\boldsymbol{y}^{\text{f}}$ are denoted by $\mathbf{H}^{\text{trans}}_{MNH}$ and $\boldsymbol{y}^{\text{f}}_{ref-MNH}$, respectively.

In order to build the linear part of the observation operator $\mathbf{H}$ and $\boldsymbol{y}^{\text{f}}$, the operator $\mathbf{H}^{\text{samp}}$ is applied. $\mathbf{H}^{\text{samp}}$ extracts the selected gradients between the monitoring sites from the simulated 4-D $CO_2$ mole fraction fields as described in Sect. 2.2. The underlying selection of the horizontal and vertical positioning of the monitoring sites in the CHIMERE grid is the same as in Bréon et al. (2015). Because the gradient selection depends on modelled wind speed and direction, the observation space $\boldsymbol{y}$ and thus $\boldsymbol{y}^{\text{f}}$ and $\mathbf{H}^{\text{samp}}$ depend on the meteorological simulations (ECMWF or Meso-NH/TEB). We denote $\mathbf{H}^{\text{samp}}$ by $\mathbf{H}^{\text{samp}}_{ini-ECM}$, $\mathbf{H}^{\text{samp}}_{ref-ECM}$, $\mathbf{H}^{\text{samp}}_{lag-ECM}$ or $\mathbf{H}^{\text{samp}}_{ref-MNH}$, depending on the inversion cases.

### 2.3.2 Emissions outside Île-de-France and model boundary conditions

$\boldsymbol{y}^{\text{f}}$ encompasses the signature of fossil fuel $CO_2$ emissions outside the Paris region but within the modelling domain, that of the modelling domain's $CO_2$ boundary conditions and that of the 30-day simulations initial conditions. The signature of the emissions outside the Paris region but within the modelling domain are simulated by CHIMERE using fossil fuel $CO_2$ emissions from the EDGAR database (Janssens-Maenhout et al., 2012). Daily $CO_2$ mole fraction fields provided by the global inversion of Chevallier et al. (2010) are used as $CO_2$ boundary conditions at the lateral and top edges of the modelling domain and as initial conditions for the $CO_2$ mole fraction fields at the beginning of each 30-day period. The global inversion of Chevallier et al. (2010) is based on the simulation of the $CO_2$ transport by the LMDZ model (Hourdin et al., 2006) and on the assimilation of ground-based measurements from a global network.

### 2.3.3 Mapping of the Paris fossil fuel $CO_2$ emissions and biogenic fluxes

$\mathbf{H}^{\text{map}}$ is built on hourly biogenic flux and emission maps at the horizontal resolution of the CHIMERE transport model. In both the initial and reference inversions, as in Bréon et al. (2015), the description of the fossil fuel $CO_2$ Paris emissions at 1- h and $2 \, \text{km} \times 2 \, \text{km}$ resolution in $\mathbf{H}^{\text{map}}$ is based on the hourly AIRPARIF 2008 inventory. The temporal profiles and spatial distributions of this inventory are analysed in Bréon et al. (2015). We just recall that emissions are available at 1 h and $1 \, \text{km} \times 1 \, \text{km}$ resolution for three typical days (weekday, Saturday, Sunday) of 5 typical months (January, April, July, August, October) of the year 2008. In order to build hourly estimates for the 1-yr period August 2010-July 2011, we follow AIRPARIF's recommendation and use January emissions for all five months from November to March, April data for all three months from April to June, and October data for both September and October and, for a given day in 2010 or 2011, we use the values from the same day in 2008.

For sensitivity tests (see Sect. 3.3.2), the emission component of $\mathbf{H}^{\mathrm{map}}$ is alternatively built based on a national emission inventory for 2005 compiled by the Institut Für Energiewirtschaft Und Rationelle Energieanwendung (IER) of the University of Stuttgart, Germany. Latoska (2009) disaggregated reported emission totals for France for 2005 into a 1×1 arc minute grid with the use of various proxies for the distribution of emitting activities such as population census, traffic intensity and land cover. We used monthly, weekly and hourly temporal profiles for different emissions sectors from the IER inventory for Europe as described by Vogel et al. (2010) to disaggregate annual emissions to hourly emissions. The IER and AIRPARIF emission inventories are two largely independent datasets.

In all experiments, the component in $\mathbf{H}^{\mathrm{map}}$ that corresponds to the biogenic control variables is based on Net Ecosystem Exchange simulated by C-TESSEL at 3 hourly and $15\,\mathrm{km} \times 15\,\mathrm{km}$ resolution. The simulated Net Ecosystem Exchange is interpolated hourly onto the CHIMERE grid (at $2\,\mathrm{km}$ to $10\,\mathrm{km}$ resolution). The C-TESSEL model does not have a specific implementation for urban ecosystems and due to its moderate horizontal resolution, it is not expected to provide a precise representation of biogenic fluxes within the urban area and in its vicinity. However, as reminded in the introduction , the signal from C-TESSEL in the $CO_2$ gradients between the peri-urban sites simulated by Bréon et al. (2015) is low. Therefore, the natural fluxes are not expected to critically affect the inversion of fossil fuel $CO_2$ emissions in our study (see forward Sect. 3). We denote $\mathbf{H}^{\mathrm{map}}$ by $\mathbf{H}^{\mathrm{map}}_{AP}$ if the hourly fossil fuel $CO_2$ flux maps are built using AIRPARIF 2008; by $\mathbf{H}^{\mathrm{map}}_{IER}$ if the hourly fossil fuel $CO_2$ flux maps are built using IER.

### 2.3.4 Building the $\mathbf{H}$ matrix

In order to apply eq. (1) and (2), $\mathbf{H}$ is built based on the different operators described above. Each column of $\mathbf{H}$ corresponds to the response of the selected $CO_2$ gradients to a control variable. Each column of this matrix is computed by applying the $\mathbf{H}$ operator (i.e. the series of operators described above) to a control vector containing only zeros except for the corresponding control variable which is set to 1.

Let $n_x$ denote the number of control variables (156 elements) for a given month of inversion, $n_f$ the dimension of the 3D flux field in the input of the CHIMERE model (i.e. the number of model horizontal grid cells times the number of hours during one month of inversion, i.e, $118 \times 118 \times 720$), $n_c$ the dimension of the 4D field of CO2 in output of the CHIMERE model (i.e. number of model grid cells times the number of hours during one month of inversion, i.e.,$118 \times 118 \times 20 \times 720$), and, at last, $n_y$ the number of gradients selected for a 1-month inversion. The dimension of $\mathbf{H}$ is $n_x \times n_y$, while the dimension of $\mathbf{H}^{\mathrm{map}}$ is $n_x \times n_f$, of $\mathbf{H}^{\mathrm{trans}}$ $n_f \times n_c$ and of $\mathbf{H}^{\mathrm{samp}}$ $n_c \times n_y$. $n_x$ application of $\mathbf{H}^{\mathrm{samp}}\mathbf{H}^{\mathrm{trans}}\mathbf{H}^{\mathrm{map}}$ are needed to build the $\mathbf{H}$ matrix. Once the $\mathbf{H}$ is built, since both $n_x$ and $n_y$ are relatively small, we can easily afford the computations in eq. (1) and (2) which involve the inversion of matrices of size $n_x \times n_x$ or $n_y \times n_y$ and multiplication of such matrices with $\mathbf{H}$.

### 2.4 Prior error covariance matrix $\mathbf{B}$

We set-up the prior error covariance matrix $\mathbf{B}$ as in Bréon et al. (2015). Assuming that there is no correlation between the uncertainties in the fossil fuel $CO_2$ emissions and the uncertainties in the biogenic fluxes, $\mathbf{B}$ is modelled as a diagonal block matrix with two blocks: one corresponds to the uncertainties in the Paris fossil fuel $CO_2$ emissions, and the other one to the Net

Ecosystem Exchange in the modelling domain. For each block, we make separate assumptions on the variance of the uncertainty in the individual control variable in one hand and on the temporal and spatial correlations between these uncertainties on the other hand.

Regarding the Paris fossil fuel emissions, we assume a 50% relative uncertainty (in terms of standard deviation) in the prior estimates of individual 6-h emission budgets. We assume that we can decompose these prior uncertainties for a given month into uncertainties in the mean diurnal cycle of the emissions and into uncertainties in the day-to-day variations of the emissions. Therefore, we compute the temporal autocorrelations of the prior uncertainties in the 6-h emission budgets, $c_t(t_1, t_2)$ (where $t_1$ and $t_2$ are two 6-h windows of 2 days of the month of inversion), as the product of the correlations of the uncertainties in the mean diurnal cycle between the four 6-h windows of the day, $c_w(w_1, w_2)$ (where $w_1$ and $w_2$ are the two 6-h windows of the day corresponding to $t_1$ and $t_2$ respectively), and correlations of uncertainty in the day-to-day variations between different days, $c_d(d_1, d_2)$ (where $d_1$ and $d_2$ are the two days corresponding to $t_1$ and $t_2$). We assume that the correlations of the uncertainty in the mean diurnal cycle between the 6-h windows of the day are positive: $c_w(w_1, w_2)$=0.4 for two consecutive windows (for example, $w_1$=0-6 h and $w_2$=6-12 h) and $c_w(w_1, w_2)$=0.2 for two non-consecutive ones (for example, $w_1$=0-6 h and $w_2$=12-18 h). The correlations of uncertainty in the day-to-day variations between different days are modelled using an exponentially decaying function with a characteristic time of 7 days: $c_d(d_1, d_2) = \mathrm{e}^{\frac{|d_2 - d_1|}{7}}$.

The standard deviation of the prior uncertainty in the 30-day budgets of Net Ecosystem Exchange for a given area and 6-h window of the day is assumed to be about 75% of the prior estimate of this budget from C-TESSEL. In practice, it appears from our computations that the resulting value of this uncertainty decreases when the surface of the corresponding area increases. Spatial and temporal correlations between the uncertainties for the various 6-h windows of the day and areas are assumed to be negligible due to the large size of the corresponding areas and due to the differences in the processes dominating the ecosystem exchanges between daytime and night-time.

## 2.5 Observation error covariance matrix R

The observation errors encompass instrumentation errors and errors in the observation operator $\mathcal{H}$. The latter combines transport model errors, representation errors, aggregation errors, errors from the boundary conditions and errors from the emissions in the modelling domain but outside the Paris area. One of the main source of transport errors is linked to errors in the wind and planetary boundary layer height in the meteorological forcing of the transport model. Representation errors are associated with the variations of $CO_2$ within the $2\,\mathrm{km} \times 2\,\mathrm{km}$ horizontal resolution grid cell of the model which encompass the peri-urban sites. They should be relatively small since there is no major $CO_2$ source in these grid cells. Aggregation errors are mainly associated with uncertainties in the spatial and temporal distribution of the emissions within the Paris area and 6-h windows. Aggregation errors are critical to account for in our inverse modelling system given that for a given 6-h window, we control one scaling factor for the emissions over the whole Paris area.

Bréon et al. (2015) used the diagnostics of Desroziers et al. (2005) to estimate the variances of the observation error. It was assumed that these errors have the same statistics for any hourly gradient and that there is no correlation between errors for different hourly gradients. Their corresponding estimate of the standard deviation of the observation error was 3 ppm.

Here, since a similar inverse modelling framework is used, and even though the revised gradient selection should decrease the aggregation errors (see Sect. 2.2), we assume that our misfits between the measured and modelled gradients bear the same observation error as in this study. $\mathbf{R}$ is thus modelled as a diagonal matrix with a $(3\,\mathrm{ppm})^2$ variance for all elements in the diagonal.

## 2.6   Principles of the sensitivity tests

Several tests are conducted to check the reference inversion results' sensitivity to changes in different components: (a) the prior estimate of the Paris fossil fuel $CO_2$ emissions, (b) the spatio-temporal distribution of the fossil fuel $CO_2$ emissions within the Paris region and within a given 6-h window, (c) the meteorological forcing driving both the atmospheric transport model and the selection of the observations. These changes are representative of typical uncertainties in these components. Most of these uncertainties are, in principle, accounted for in the configuration of the prior uncertainty covariance matrix $\mathbf{B}$ and observation error covariance matrix $\mathbf{R}$, respectively. Their impact on the robustness of the inversion results should be given by the posterior uncertainty covariance matrix $\mathbf{A}$. However, they may not be correctly reflected by the statistical representation that is based on Gaussian and unbiased distributions and by the rather simple models used to set up the covariance matrices. Therefore, the sensitivity tests provide a useful alternative evaluation of the robustness of the inversion results.

Regarding the prior estimate of the Paris fossil fuel $CO_2$ emission budgets, as an alternative to the AIRPARIF 2008 budgets, we use what is called hereafter flat priors, i.e., prior fossil fuel $CO_2$ emission estimates that are not informed about month to month variations. Three sets of flat priors are built by rescaling the AIRPARIF 2008 budgets using monthly, daily or 6-h scaling factors. In the first case, the flat priors have constant monthly values, but retain the relative temporal variations of the 6-h budgets within a month. In the second case the flat priors have constant daily values, but retain the relative temporal variations of the 6-h budgets within a day. In the third case, the flat priors have constant 6-h mean values. This change of prior estimate can potentially have a large impact on the results since the system assimilates data during the afternoon only. Consequently, such a change imposes a direct constraint on two 6-h windows of the day only (the 6-12 h and the 12-18 h windows) while the constraint on the two other windows (the 0-6 h and the 18-24 h windows) relies indirectly on the description of the temporal correlations in the prior uncertainty covariance matrix $\mathbf{B}$.

For each set, different flat priors are tested by taking different values for the monthly budgets. These values cover a case of relatively high emissions ($5\,\mathrm{MtCO_2 month^{-1}}$), a case of relatively low emissions ($3\,\mathrm{MtCO_2 month^{-1}}$), as well as an intermediate case corresponding to the annual budget from the AIRPARIF 2008 inventory ($4.3\,\mathrm{MtCO_2 month^{-1}}$). A prior estimate based on the budgets from the IER inventory is also used for the sensitivity tests. As explained in Sect. 2.3.3, $\mathbf{H}^{\mathrm{map}}_{IER}$ is used as alternative $\mathbf{H}^{\mathrm{map}}$ to $\mathbf{H}^{\mathrm{map}}_{AP}$, while $\mathbf{H}^{\mathrm{trans}}_{ref-MNH}$, $\boldsymbol{y}^{\mathrm{f}}_{ref-MNH}$ and $\mathbf{H}^{\mathrm{samp}}_{ref-MNH}$ are used as alternative $\mathbf{H}^{\mathrm{trans}}$, $\boldsymbol{y}^{\mathrm{f}}$ and $\mathbf{H}^{\mathrm{samp}}$ to $\mathbf{H}^{\mathrm{trans}}_{ECM}$, $\boldsymbol{y}^{\mathrm{f}}_{ref-ECM}$, $\mathbf{H}^{\mathrm{samp}}_{ref-ECM}$. Table 1 summarizes the acronyms and settings of the different sensitivity tests.

## 3 Results

Bréon et al. (2015) analysed the skill of the inversion by comparing the fit between measured and modelled $CO_2$ gradients, which is a first indicator of the reliability of the inverted emissions. In Tab. 2, statistical comparisons between the selected measured gradients and results from the initial and reference inversion, respectively, are provided. It demonstrates that both

inversions strongly increase the consistency between model and measurements compared to the prior simulations (Tab. 2). Of note is that the statistics of both the prior and posterior model-data misfits are smaller for the initial inversion than for the reference inversion. This is explained by the fact that the initial inversion selects gradient for which the signal (mainly the impact of the city emissions) is smaller than for the gradients that both inversions select. However, we avoid a more detailed analysis of the model-data misfit in the following.

Using the loose wind ranges of the initial inversion to select gradients, the root mean square of the biogenic signal in these hourly gradients, averaged over the 1-yr CO2-MEGAPARIS period, is $1.1 \, \mathrm{ppm}$, which is, as indicated in introduction, much lower than the signal from the Paris emission. When using the tighter wind ranges of the reference inversion, the root mean square of the biogenic signal in these gradients is even smaller ($0.8 \, \mathrm{ppm}$). Therefore, the changes in the inversion configuration proposed in our study decrease the impact of the uncertainty in the natural fluxes. This weak impact was already demonstrated

by Bréon et al. (2015). We thus do not analyse further the results that correspond to the control of the natural fluxes in the following.

     The presentation of the results rather focuses on the estimates of the monthly fossil fuel $CO_2$ emission budgets from mid-2010 to mid-2011, expressed in $\mathrm{MtCO_2 month^{-1}}$ (strictly speaking $\mathrm{MtCO_2}$ per 30 days, see Sect. 2). The uncertainties (in terms of standard deviation) in prior and posterior estimates of the monthly emissions are based on the modelling of the $\mathbf{B}$

matrix, described in Sect. 2.4, and on the derivation of the $\mathbf{A}$ matrix (eq. (2)). However, the robustness of the results is evaluated using independent knowledge on the emissions rather than using these theoretical indicators. AIRPARIF (2013) reports an estimate of $41.8 \, \mathrm{MtCO_2 yr^{-1}}$ for the annual emission budget of Île-de-France in 2010. This number is used as an indicator for the evaluation of the 1-yr budget of the estimates from the inversion. According to AIRPARIF (2013), the residential and the service sector account for 43% of the Paris fossil fuel $CO_2$ emissions in 2010. These emissions are almost entirely linked

to heating. Heating in the industry sector contributes also significantly to Paris' emissions. The heating is mostly dedicated to ambient air in the buildings and to sanitary water. Therefore we expect a large increase of the emissions from summer to winter and a high correlation between these emissions and the temperature during the cold season. An independent analysis of both daily gas use and hourly electric consumption within Île-de-France indicate a heating energy use that is highly correlated to the daily-mean temperature when this temperature is below $19^o \, \mathrm{C}$, and essentially independent of the daily mean temperature

when this temperature is above $19^o \, \mathrm{C}$ (unpublished analysis led by one co-author of this study, François-Marie Bréon). For the evaluation of the results, our emission estimates are thus compared with monthly averages of an independent measure, which we call heating degrees hereafter. It is defined as the positive difference between the daily mean temperature and $19^o \, \mathrm{C}$ (set to 0 for days when the temperature is higher than $19^o \, \mathrm{C}$). The ratio between January and July emission estimates from the AIRPARIF 2008 inventory seem surprisingly low given these considerations. Furthermore, the prior estimates based on this

inventory make use of a single emission value from November to March, which does not account for the large temperature variations during this period. Therefore, an amplified seasonal cycle of the emissions that better correlates with heating degrees is expected through the atmospheric inversion.

## 3.1 Initial inversion using loose wind direction criteria for the gradients selection

Figure 3a shows prior and posterior estimates of the Paris emissions from the initial inversion (experiment *ini*, Tab.1). Monthly mean heating degrees for the centre of Paris, derived from the temperature given by the ECMWF's operational analysis, are also shown on this figure. The posterior estimates are lower than the prior ones for all months. The inversion decreases the annual emissions from $51.9\,\mathrm{MtCO_2yr^{-1}}$ to $37.4\pm2.1\,\mathrm{MtCO_2yr^{-1}}$. This number is smaller but closer to the AIRPARIF inventory 2010 used for evaluation ($41.8\,\mathrm{MtCO_2yr^{-1}}$, see Tab. 1).

Posterior fluxes are lowest for August ($1.6\,\mathrm{MtCO_2month^{-1}}$), increase steadily until peaking in February ($4.6\,\mathrm{MtCO_2month^{-1}}$), drop strongly in March ($2.8\,\mathrm{MtCO_2month^{-1}}$), and vary between 2 and $3\,\mathrm{MtCO_2month^{-1}}$ from April to July. Compared to the prior estimate, the inversion yields larger emissions in winter and increases the amplitude of the seasonal variations. For the period analysed, monthly mean heating degrees were highest in December (Fig. 3a). Hence, fossil fuel $CO_2$ emissions are expected to be the highest in December rather than in February. There is no clear correlation between monthly heating degrees and emission estimates during the November-March period.

The number of assimilated gradients varies considerably from one month to another, which influences the month-to-month variations of the inverted emissions. For instance, 163 observations are assimilated in March, compared with only 34 in November. Figure 3a also shows that, for most months, the numbers of selected gradients are not apportioned equally amongst the NE and SW wind directions. For instance, there are no NE gradients to constrain August emissions, while less than half of the gradients in March are SW gradients. The different upwind conditions for NE and SW gradients could play a role in the month-to-month variability of the inverted emissions, in case the gradient approach does not remove entirely the influence of remote fluxes.

We investigate the impact of assimilating these two different gradient types on monthly fossil fuel $CO_2$ flux estimates by conducting inversions based on NE gradients, SW gradients, or even GIF-MON, GIF-GON, MON-GIF or GON-GIF only (Fig. 3b). The difference in inverted December emissions when assimilating only SW gradients compared to assimilating only NE gradients is large, even though a large number of both types of gradients is available during this month. Compared to the prior estimate, the inversion of SW gradients increases the December emissions. The opposite, however, is true for the inversion of NE gradients. This behaviour seems to be driven by both the assimilation of GIF-MON and GIF-GON gradients. An analysis of the average temperature in Paris (not shown) shows lower temperatures for NE wind conditions than for SW wind conditions. The heating emissions in Paris should thus be higher for NE wind conditions. Therefore, the temperature variations cannot explain the differences in December emissions between assimilating SW gradients and assimilating NE gradients.

The differences between the results when using NE gradients or SW gradients are not as large for other months as in December. However, they can still be significant, e.g., April (Fig. 3b). These differences cannot be explained by a lack of data

for a given type of gradient, except in August, when there are no NE gradients. For January and February, differences of 1 to 1.5 $MtCO_2$ (i.e., about 35 % relative differences) are obtained between inversions assimilating only MON-GIF compared to assimilating only GON-GIF, although both are SW gradients and gather more than 40 observations during each month. This large mismatch between the different inversions when using different data subsets undermines the reliability of the inversion

results, in particular in December. The seasonal profile of the retrieved emission when assimilating only SW gradients is far better correlated with heating degrees than when the inversion uses both SW and NE gradients, as emissions reach their maximum in December. Results seem nearly as sensible when using MON-GIF or GON-GIF as when using all SW gradients.

      Figures 4a and 4c illustrate that, even if we discard GON-MON and increase the threshold of the wind speed, there are episodes when measured gradients show negative values. They, however, should be positive owing to the city's emissions.

Most of the negative gradients are found when the wind is from NE, such as in December (Fig. 4a), suggesting that such gradients may not represent the emissions of the entire city.

### 3.2   Reference inversion

The estimates of the monthly Paris fossil fuel $CO_2$ emissions by the reference inversion (experiment *ref*, Tab.1) are given in Fig. 3c. All negative observed gradients outlined above were obtained at the limit of the wind direction range proposed by

Bréon et al. (2015). As illustrated for December and May by comparing Fig. 4a to Fig. 4b and Fig. 4c to Fig. 4d, respectively, the reference inversion, which uses a stricter range of wind directions (Sect. 2.2), removes the negative gradients. Despite the loss of 65 % of the data compared to the initial inversion, the reference inversion still predicts a large uncertainty reduction for monthly fossil fuel $CO_2$ emission estimates, from 9 % in November to 50 % in October.

      The Paris monthly fossil fuel $CO_2$ emission estimates from the reference inversion correlate well with monthly heating

degrees ($r^2$ = 0.67 for the whole period, $r^2$ = 0.45 for November–February), which was not the case of the initial inversion ($r^2$ = 0.54 for the whole period, $r^2$ = 0.07 for November–February). In general, the reference inversion decreases the fossil fuel $CO_2$ budget from the prior estimate, except in December, which becomes the peak of emissions (Fig. 3c). The emissions decrease from February to March, which does not correspond to a relative change in heating degrees, is significantly smaller in the reference than in the initial inversion. The seasonal variations of the reference inversion are strongly improved compared

to initial inversions. The annual budget from the reference inversion (40.9 $MtCO_2$) is close to that from AIRPARIF 2010 (41.8 $MtCO_2$, see Tab. 1).

      The stricter gradient selection further leads to a much better agreement between emission estimates when using different subsets of gradients (compare Fig. 3d with Fig. 3b). Although significant differences in December and April emissions are still apparent between using NE gradients or using SW gradients, and in January and February emissions between using GON-

GIF or MON-GIF, these differences are smaller than in the initial inversion. Now, even when assimilating only NE gradients, the four months with largest inverted emissions correspond to the four coldest ones with the highest heating degrees of the year (November to February), though the assimilation of NE gradients still leads to smaller emissions in December than in November, January and February. One may argue that the improvements of the reference over the initial inversion reflect the assimilation of a smaller dataset, and therefore are due to smaller corrections. However, results from the reference and initial

inversion are closer to each other if only SW gradients are assimilated than if only NE gradients are assimilated. Highest correlation with the heating degrees are obtained when only SW gradients are assimilated.

The theoretical posterior uncertainties in the monthly budgets of the emissions are generally much lower than the prior uncertainties (with more than 30% uncertainty reduction for most of the months) in the reference inversion. However, even though our analysis above gives more confidence in the results from the reference inversion than in the results from the initial inversion, the inverse modelling system diagnoses smaller posterior uncertainties in the latter than in the former.

Results from the inversion using a 2- h lag between the upwind and the corresponding downwind measurement are shown in Fig.5. The corrections applied to the prior estimate from AIRPARIF 2008 by this inversion are qualitatively consistent with those of the reference inversion. The amplitudes of the corrections, however, are much smaller as the large decrease of the number of assimilated gradients in this inversion compared to the reference configuration (only 4% of available measurement are used by reducing the time window of eligible upwind or downwind measurements; see Sec.2.2) clearly limits the weight of the observational constraint.

### 3.3 Sensitivity Tests

#### 3.3.1 Sensitivity of the Paris emission budgets to prior estimates $x_b$

Monthly fossil fuel $CO_2$ emissions estimates using flat priors for $x_b$ (all experiments *FLAT_* in Tab. 1) are reported in Fig. 6. Although differences in prior monthly budgets between the FLAT_3.0, FLAT_4.3 and FLAT_5.0 experiments amount to $2\,MtCO_2month^{-1}$, posterior differences between their monthly budgets are generally much lower. In addition, the posterior monthly emissions when using flat priors are comparable to those from the reference inversion—with a very similar month-to-month variation. The differences between posterior monthly emissions from the FLAT_$m$M and FLAT_$m$D ($m$=3.0, 4.3 or 5.0) inversions and the reference inversion are generally smaller than $1\,MtCO_2month^{-1}$, except during September, November, May and July when very few (4 to 24) gradients are assimilated (Fig. 6a and Fig. 6b). Larger differences are obtained between the reference inversion and FLAT_$m$H ($m$=3.0, 4.3 or 5.0) inversions, which use prior estimates that are flat at 6- h scale ( Fig. 6c). The FLAT_$m$H experiments yield larger posterior monthly budgets than the FLAT_$m$D and FLAT_$m$M experiments.

Posterior annual budgets from FLAT_$m$M and FLAT_$m$D inversions range between 33 and $45.3\,MtCO_2yr^{-1}$, encompassing the budgets from the reference inversion and AIRPARIF 2010 (Tab. 1). In particular, the inverted annual budget from FLAT_4.3M and FLAT_4.3D, whose prior estimate have the same annual budget as the prior estimate from the reference inversion, is equal to $41.1\,MtCO_2$. This is very close to the annual budgets from the reference inversion and AIRPARIF 2010. However, the annual emissions budgets from the FLAT_$m$H inversions range from 33 to $52.2\,MtCO_2$, which is biased compared to both the reference inversion and AIRPARIF 2010.

#### 3.3.2 Sensitivity of the Paris emissions budgets to the mapping and variations at hourly scale

Figure 7 compares the estimates from the reference inversion, which uses $\mathbf{H}^{map} = \mathbf{H}^{map}_{AP}$, to the estimates from the sensitivity test with $\mathbf{H}^{map} = \mathbf{H}^{map}_{IER}$ (INV_mapIER and INV_IER, see Tab. 1). Thus, this experiment also includes results when using the

IER inventory to build both the 6-h budgets in $\boldsymbol{x}_b$ and $\mathbf{H}^{\mathrm{map}} = \mathbf{H}_{IER}^{\mathrm{map}}$ (INV_IER). It provides estimates when the inversion relies entirely on the IER inventory to define these parameters and ignores the existence of the AIRPARIF inventory. This situation is similar to that in cities, where no local inventory is available. We have less confidence in the posterior estimates from such an inversion, since the IER inventory does not rely on the same amount of local data as the AIRPARIF inventory.

5    INV_mapIER regularly predicts lower monthly budgets than the reference inversion, except in June, July, September and November. The corresponding differences are relatively small and do not exceed $0.5\,\mathrm{MtCO_2 month^{-1}}$, except in January and February. Similar to the reference inversion, INV_mapIER predicts highest emissions in December. However, its estimates for January and February fluxes are particularly low, e.g., January estimates ($3.9\,\mathrm{MtCO_2 month^{-1}}$) roughly equal that for May ($3.9\,\mathrm{MtCO_2 month^{-1}}$) or October ($3.8\,\mathrm{MtCO_2 month^{-1}}$), and are smaller than that for September ($4.1\,\mathrm{MtCO_2 month^{-1}}$). 10    This results in an annual budget of $39\,\mathrm{MtCO_2 yr^{-1}}$ that is still closer to the one from AIRPARIF 2010 than to the one from AIRPARIF 2008 (Tab. 1).

The monthly prior emissions from AIRPARIF 2008 and IER differ substantially. In particular, from November to May, the IER inventory estimates up to $3\,\mathrm{MtCO_2 month^{-1}}$, (approximately 40%) higher fossil fuel $CO_2$ emissions for the Paris region than AIRPARIF 2008. At the annual scale, estimates differ by $8.2\,\mathrm{MtCO_2 yr^{-1}}$ (Tab. 1). The differences between the 15    two inventories are due to both the differences in the emission model and the driving activity data used. The two inventories correspond to two different years (2008 versus 2005). However, this hardly explains the amplitude of the difference between the two inventories by itself. The decrease of the total emission in France between 2005 and 2008 was approximately $5\,\%$ (see, e.g., CITEPA, 2015). Here, the difference in total emissions in Île-de-France between the IER and AIRPARIF 2008 inventory, however, is about 14%. Results of the inversion using IER for both the prior emission budgets and the emissions' 20    spatial distribution (INV_IER) reflect these large prior discrepancies. Indeed, monthly and annual budgets of Paris' fossil fuel $CO_2$ emissions estimated by INV_IER are larger than that from the reference inversion and from INV_mapIER (Fig. 7). The differences in posterior February emissions from IER_INV and the reference inversion exceed $2\,\mathrm{MtCO_2 month^{-1}}$. The discrepancies are even larger, when comparing the monthly emission estimates from INV_mapIER and IER_INV, since the change of $\mathbf{H}^{\mathrm{map}}$ from $\mathbf{H}_{AP}^{\mathrm{map}}$ to $\mathbf{H}_{IER}^{\mathrm{map}}$ has a tendency to decrease the posterior emission estimates.

25    The IER inventory indicates higher emissions in March than in November. Posterior estimates from INV_IER still indicate that the highest emissions are in November-February. Due to a residual influence from the IER prior estimate, INV_IER predicts highest emissions in February. The December emission estimate is close to February emission estimate, and is the second highest 1-month mean estimate from INV_IER. Finally, the annual posterior emission from INV_IER is closer to that from AIRPARIF 2010 than to that from the 2008 inventory, despite the far higher prior annual estimate from the IER inventory 30    (Tab. 1).

### 3.3.3   Sensitivity to $\mathbf{H}^{\mathrm{samp}}$, $\mathbf{H}^{\mathrm{trans}}$ and $y^{\mathrm{f}}$

INV_MNH is compared to the reference inversion to analyse the impact of using Meso-NH/TEB instead of ECMWF as meteorological simulation for both the gradient selection in the observation vector ($\mathbf{H}^{\mathrm{samp}} = \mathbf{H}_{ref-MNH}^{\mathrm{samp}}$ versus $\mathbf{H}^{\mathrm{samp}} = \mathbf{H}_{ref-ECM}^{\mathrm{samp}}$) and the forcing of CHIMERE ($\mathbf{H}^{\mathrm{trans}} = \mathbf{H}_{MNH}^{\mathrm{trans}}$ versus $\mathbf{H}^{\mathrm{trans}} = \mathbf{H}_{ECM}^{\mathrm{trans}}$ and $y^{\mathrm{f}} = y_{ref-MNH}^{\mathrm{f}}$ versus $y^{\mathrm{f}} = y_{ref-ECM}^{\mathrm{f}}$). Meso-

NH/TEB data are available up to June only which explains that the analyses here are restrained to the period August 2010 to June 2011.

The time series of the gradients that are selected for the assimilation using $\mathbf{H}_{ref-MNH}^{samp}$ and $\mathbf{H}_{ref-ECM}^{samp}$, respectively, are shown in Fig. 2 (and Fig. A1–A2). The significant differences in selected gradients apparent in Fig. 2 (and Fig. A1–A2)) are driven by small differences in simulated wind fields between ECMWF and Meso-NH/TEB. Small differences in wind direction and speed are often sufficient to cross the thresholds defining the gradient selection (Fig. A1–A2). This differences result in a significantly different set of assimilated gradients and in a different apportionment according to prevailing NE or SW wind directions (Fig. 8).

Despite this, Fig. 8 reports similarities in inverted monthly emissions from INV_MNH and the reference inversion. Differences in monthly posterior emission estimates are less than $0.5\,\mathrm{MtCO_2 month^{-1}}$ when assimilating all selected gradients (Fig. 8). The four highest emitting months are still November to February for INV_MNH. However, larger differences to the reference inversion estimates are found for December and May, resulting in the loss of the peak in December and in an unexpected peak in May in INV_MNH (Fig. 8a). This disagreement is related to the assimilation of NE gradients. As shown in Fig. 8b, emissions estimates from INV_MNH and the reference inversion are very similar when only SW gradients are assimilated. By contrast, large differences are obtained in December and May when only NE gradients are assimilated (Fig. 8c). The larger fraction of selected NE gradients compared to selected SW gradients when using Meson-NH/TEB instead of ECMWF could explain the loss of the emission peak in December. There is no peak in December when using either Meso-NH/TEB, or ECMWF and NE gradients only. Nevertheless, when assimilating SW gradients, the consistency between INV_MNH and the reference inversion is surprising, given the significantly different SW gradient selection.

## 4  Discussion and Conclusions

### 4.1  Summary and general analysis of the results

We have analysed estimates of monthly mean fossil fuel $CO_2$ emissions from the Paris urban area from August 2010 to July 2011 using continuous $CO_2$ measurements from three stations and a city-scale atmospheric inverse modelling framework derived from Bréon et al. (2015). The inversion modelling is based on a mesoscale configuration of CHIMERE, on the AIRPARIF high-resolution $CO_2$ emission inventory for 2008, on the C-TESSEL simulation for the biogenic fluxes in Northern France, and on the principle of constraining the 6-h city-scale budget of the emissions using cross-city $CO_2$ gradients. As demonstrated by the analysis of the inversion results, this study has critically improved configuration of Bréon et al. (2015) by (i) discarding GON-MON gradients since that are not related to the emission of the entire city, and (ii) by using stricter criteria on the wind direction and wind speed for the selection of gradients.

The analysis suggests an improvement of the city's seasonal to annual emission budget from the reference inversion compared to the prior estimate that is based on the AIRPARIF 2008 inventory. The inversion derives an annual emission budget (for August 2010–July 2011) of $40.9\,\mathrm{MtCO_2 yr^{-1}}$, which is closer to the independent estimate from the AIRPARIF 2010 inventory ($41.8\,\mathrm{MtCO_2 yr^{-1}}$) than to the prior estimate ($51.9\,\mathrm{MtCO_2 yr^{-1}}$). Although the reported estimate from the AIRPARIF 2010

inventory does not exactly correspond to the mid-2010 to mid-2011 period, changes between the 2008 and 2010 inventories reflect improvements in the inventory model and actual changes of the Paris emissions. Therefore, the fact that the corrections applied by the inversion to the prior estimate from AIRPARIF 2008 are consistent with the differences between the AIRPARIF 2008 and 2010 inventories gives confidence in the inversion.

The seasonal variations of the monthly inverted emissions also appear more realistic than that of the prior emission estimates. The seasonal amplitude of the emissions revealed by the reference inversion is higher than that of the prior estimate of the emission, derived from the 5 typical months of the AIPARIF 2008 inventory. This increase in amplitude makes sense, given that a large fraction (43 % according to the AIRPARIF 2008 inventory) of the Paris emissions are due to domestic and commercial heating. It is supported by the fact that the seasonal variations in the AIRPARIF 2010 inventory are higher than that derived

from the AIRPARIF 2008 inventory. The inverted seasonal cycle of the emissions correlates well ($r^2$ =0.45) with the heating degrees in fall-winter (November–February). The four months with highest inverted emissions correspond to the four coldest months (November to February)–with a peak in both the emissions and the heating degrees in December. By contrast, the prior estimate of the emissions derived from AIRPARIF 2008 does not differentiate monthly budgets from November to March.

The sensitivity tests indicate that the uncertainties assigned to the prior estimates of the 6- h mean emissions, to the spatio-

temporal distribution of the emissions within the Paris area and 6- h windows, and to the meteorological simulations (for the cross-city gradient selection and for the forcing of CHIMERE) have a moderate impact on the monthly mean emission estimates once the inversion is driven by the most stringent selection of the measurements. This weak sensitivity of the inverted emissions to the uncertainties assigned to the inverse modelling components is important for the credibility of the inversion approach in view to apply this approach as an independent method to verify inventories. Here, the inverted emission budgets

are sensitive to each of the above-mentioned components. However, even though we assimilate a relatively small number of data, this sensitivity is generally much smaller than the differences between inverted and prior estimates at monthly to annual scale. Furthermore, the plausible seasonal variations of the emissions revealed by the reference inversion is robust to most sensitivity tests.

The inversions generally return smaller emissions than the prior estimates. This is even the case when using a prior estimate

that is flat at the monthly scale only, and that has an annual emission budget of $36\,\mathrm{MtCO_2yr^{-1}}$, i.e., a budget that is smaller than that from AIRPARIF 2010. The inversion decreases the annual emission budget when using the diurnal cycle of the emissions from AIRPARIF 2008 as prior estimate of the 6- h mean emissions. In contrast, the annual emission budget is increased when a flat diurnal cycle and a prior estimate of the annual emission budget that is smaller than that from AIRPARIF 2010 (i.e. of $36\,\mathrm{MtCO_2yr^{-1}}$) is used. This can reveal an error in the mean diurnal cycle of the emissions from AIRPARIF

2008, which the inversion could not correct for since data are assimilated during afternoon only. Moreover, we define the uncertainties in the prior emission estimates in terms of relative rather than absolute uncertainty. Consequently, using the diurnal cycle of the emissions from AIRPARIF 2008 in the prior estimate of the 6- h mean emissions and higher (smaller) prior emissions at monthly to annual scale leads to higher (smaller) prior uncertainties, and thus to a stronger (weaker) constraint from the atmospheric measurements, resulting in a stronger (weaker) decrease of the emissions. One could argue that this

artificially helps getting a robust convergence of the sensitivity tests using different prior estimates and it likely plays a role at

the annual scale. This could be problematic, since having a fixed value for the relative uncertainty in the prior estimates is not suitable when these estimates become very small. However, for some months, the convergence between inversions utilising different flat priors is obtained by both positive and negative corrections. This is the case in January and February 2011 for FLAT_$m$M experiments (Fig.6a). The convergence can also be obtained with positive corrections that are larger when prior

uncertainties are smaller, e.g., in December 2010 for FLAT_$m$M experiments (Fig.6a). Figure 6c gives several examples where the monthly budget of the prior estimate that are flat at the 6- h scale determines if the corresponding corrections are positive or negative. This figure also illustrates the fact that the amplitude of the correction to the monthly estimates is not highly correlated with the corresponding prior uncertainty. Furthermore, the fact that higher prior emission estimates are assigned higher prior uncertainties cannot explain the level of convergence of the sensitivity tests. In particular, it can not explain the robustness

of the retrieved seasonal cycle of emissions when using flat priors. It neither explains the fact that the annual budget from INV_IER is closer to AIRPARIF 2010 than to AIRPARIF 2008. INV_MNH selected a significantly different set of gradients. However, it still constrains the inverted emissions towards the same levels of emissions as in the reference inversion (typically differences in monthly emissions are <5 %).

     The improvement of the reference inversion compared to the initial inversion demonstrates the need for a narrow definition

of the wind direction ranges, and more generally the need for a very careful selection of $CO_2$ data. This reveals the asset of following as much as possible the concept of assimilating cross-city $CO_2$ gradients to control the emissions at the whole city scale, and to filter out the poorly modelled influence of fluxes outside the Paris urban area. The assimilation of gradients cannot perfectly cancel this influence because firstly one cannot set up the inversion system to ensure that the selected gradients correspond to the concentration variations of air masses that travel from the upwind to the downwind sites (at least due to

uncertainties in the atmospheric transport) and secondly because the signal from fluxes outside the Paris varies during such a transport (due to atmospheric diffusion). However, results from Bréon et al. (2015) and from this study demonstrate that the assimilation of gradients succeeds in decreasing this signal. These studies also show that such a decrease strengthens the inversion results by limiting the problem of the uncertainties in the remote fluxes for regional inversions (which is particularly critical in the Paris area as shown by Bréon et al., 2015) and the problem of the uncertainties in natural fluxes for urban $CO_2$

emission inversions. The positive insights from the evaluation of our results also strengthens the confidence in this relatively simple concept to estimate monthly budgets of the city emissions, even if it relies on the assimilation of a relatively small amount of data.

## 4.2    Problems to be solved

The different inversion tests still raise concerns for the inversion of the cities' monthly emission budgets. We expected that

cross-city gradients would be weakly sensitive to the uncertainties in the distribution of the emissions within the Paris region and the 6- h windows, which explains why we control, for a given 6- h window, a single scaling factor for the emissions of the entire urban area. The inversion results, however, are significantly affected by changes in the emission distribution. This does not necessarily question the control of a single scaling factor for the emissions of the whole urban area since reasonable results

are obtained using the emission distribution from AIRPARIF but it reveals the need to rely on robust, high resolution emission maps such as those produced by local agencies like AIRPARIF. However, many cities do not have such local inventories.

Bréon et al. (2015) have shown that the selection of afternoon data provides little constraint on night-time emissions. This is problematic since the diurnal cycle is highly uncertain in inventories. The differences between the results from FLAT_$m$D and FLAT_$m$H indicate that the poor representation of the diurnal cycle in FLAT_$m$H has a large impact on the inverted monthly emissions. As highlighted above, the inversions, based on the diurnal cycle from AIRPARIF 2008, generally tend to decrease the prior estimates which can also be viewed as an impact from errors in this diurnal cycle. New approaches and techniques are needed to provide a direct constraint on the night-time emissions or to better extrapolate the information from daytime to night-time data to solve for this problem. The poor representation of the day-to-day variations in the flat prior of FLAT_$m$D does not seem to impact the results from this inversion, which are close to that of FLAT_$m$M. Even though there is a large number of days, even sometimes weeks, during which no gradients are assimilated, the inversion does not strongly rely on the prior day-to-day variations within the months to correct the monthly mean emissions budgets. However, there is a critical lack of data, which is primarily due to the small number of sites available for this study, and thus to the relatively small wind sectors by which we select cross-city gradients. This lack of data hinders the results of all inversions for specific months such as September, November, April, May and July, when less than 30 1-hour mean gradients are assimilated. The month-to-month variability is thus often driven by the variability of the data availability. Results at the monthly scale are thus not systematically consistent with the different sensitivity tests. Monthly estimates can be weakened by missing or over-weighting high variations in the emissions over short time periods (e.g., due to a cold event). One can hope that this limitation could be overcome by an expansion of the observation network with stations all around the Paris urban area, which could ensure a continuous monitoring of the cross-city $CO_2$ gradients.

In December, the number of assimilated data is relatively high for both the reference inversion and INV_MNH. However, while the inversions increase the emissions compared to AIRPARIF 2008 during December when using ECMWF data and all gradients (SW and NE gradients), this is not the case when assimilating subsets of the cross-city gradients only, or when using Meso-NH/TB. Consequently, there is no peak of inverted emission estimates in December. Neither is this a robust feature of the reference inversion. The absence of an emission peak in December is associated with the assimilation of NE gradients (i.e., due to the assimilation of NE gradients only, or, to the use of the Meso-NH/TEB meteorology which selects a larger fraction of NE gradients than its ECMWF counterpart).

More generally, the assimilation of NE gradients seems to raise concerns while more satisfying results are obtained when using SW gradients. This applies also to the initial inversion, for which the NE direction corresponds to wider wind direction ranges. Thus, the problem cannot be related to a very specific source NE of Paris. When the wind blows from NE, the signature of emissions from remote, highly urbanised and industrialised areas (North-Eastern France, Benelux and Western Germany) should impact the $CO_2$ fields in the Paris area. On the opposite, the regions between the Atlantic Ocean and Paris are mostly rural. While the computation of gradients is an efficient way of limiting the signatures of the fluxes outside the Paris area on assimilated data, and while it effectively reduces these signatures to a small component in the simulated gradients, it does not ensure a total removal of such signatures in the measurements which may bear a more spatial heterogeneity than the modelling

framework. The large-scale signature of the remote natural fluxes from SW may be more easily modelled or filtered out by the computation of gradients in the Paris area than the signature of emissions from NE. This could explain why the assimilation of NE gradients is more problematic than that of SW gradients. This could reveal another limitation in assimilating cross-city gradients. The high temporal variability of the ratio of assimilated NE gradients to SW gradients may be problematic for the monitoring of the month-to-month variability of the city emissions. Similarly, the small biogenic signal in the simulated gradients may be due to the use of an ecosystem model with moderate horizontal resolution. Measured gradients might be impacted by urban ecosystems that cannot be represented with this model. Due to the high density and compactness of the Paris urban area, we can assume that such urban ecosystems should have a low impact on the inversion of Paris emissions. This should be further investigated based on urban ecosystem modelling and monitoring (Nordbo et al., 2012).

The last major issue is the limited confidence in the posterior uncertainties computed by the inversion system. We purposely avoided analysing them in details in Sect. 3. They provide qualitative insights on the behaviour of the inversion, i.e., posterior uncertainties remain close to the prior ones for night-time emissions, which are poorly constrained by using only afternoon $CO_2$ data (Bréon et al., 2015). The posterior uncertainties also vary as a function of the number of assimilated data. The different estimates from the sensitivity tests generally lie in the 68% confidence (1-$\sigma$) interval of the reference inversion. However, the posterior uncertainties look generally very low for specific months, despite the lack of confidence in the specific monthly estimates as discussed above, and despite the very limited number of assimilated data. During February and March, the posterior uncertainties from the reference inversion are lower than $0.69\,\mathrm{MtCO_2yr^{-1}}$. The large emission decrease of $1.32\,\mathrm{MtCO_2yr^{-1}}$ from February to March is surprising. The relative difference between the posterior and prior uncertainties when moving from the initial inversion to the more reliable reference inversion demonstrates how misleading the interpretation of theoretical uncertainties can be when several mathematical assumptions in the inversion are not met in practice. However, even though the configuration is far from perfect, the misfits between posterior estimates and observations are still smaller than between prior estimates and observations. This gives a stronger confidence in posterior emission estimates than in the posterior uncertainties of these emissions. Sensitivity tests with the analysis of the posterior estimates only were conducted to give a better picture of the strength of the measurement constraint.

## 4.3 Perspectives

Despite these concerns, the results from this study are promising and several methodological improvements were found. The inversion test of assimilating spatio-temporal gradients accounts for the time air parcels need to pass from the upwind site over the urban area to the downwind site. Such gradients should bear a smaller signal from fluxes outside the urban area than spatial gradients, which should help isolating this signal from the city emissions. The lack of data, however, prevented this inversion from significantly departing from its prior emission estimate. Such a strategy would be more appropriate if a larger amount of data was available, but it is impractical for our limited network: it exacerbates the loss of data from already strict gradient selection criteria and degrades the overall emission retrieval compared to the reference inversion. For the same reason, it would be inappropriate with our limited network to narrow the wind direction ranges to select gradients to less than $\pm 15^o$ of the

transect between the downwind and upwind sites, even though, in principle, it would strengthen the decrease of the signature from fluxes outside the urban area.

The expansion of the network, in particular a full encirclement of the city with at least 8 sites (given that the wind ranges for the selection of gradients between one upwind site and one downwind site cover 30 degrees in this study) should strengthen the results and could allow applying such new techniques that result in a stricter gradient selection. However, relying on such a measurement expansion may not be sufficient. Exploiting more information from the available dataset without violating or undermining our assumptions on the selection of cross-city gradients is a requirement to strengthen the observational constraint of the inversion. The Paris observation network has been set back since September 2014 in the framework of the CARBOCOUNT-CITY and LE CO2 PARISIEN projects. Both projects aim to deploy more measurement sites than the CO2-MEGAPARIS project. However, relying on such a measurement network expansion may not be sufficient. New methods should be developed to exploit urban measurements (Wu et al., 2016) which would allow to solve for the spatial distribution of the emissions, which does not seem possible with the current monitoring network of peri-urban sites. This in turn could help assimilating data that do not necessarily bear the signature of the emissions from a large part of the city. Finally, developing methods to exploit morning, evening and night-time data would be necessary to constrain night-time fluxes. This is not necessary to improve the knowledge on the emissions based on atmospheric inversion, but this is necessary to develop accurate tools for the operational monitoring and verification of the emissions based on this approach.

Even though it applies to the specific case of monitoring the $CO_2$ emissions from Paris, this study demonstrates the potential of an approach which can be adapted to a wide range of cities. The urban surrounding, spread, size, topography and meteorology of some cities increase the difficulty for catching cross-city gradients, and different strategies may be more adapted for such cases. The atmospheric inversion of the city emissions is still an emerging activity, but the present results already raise some confidence in this concept, especially since many other resources (combining atmospheric $CO_2$ inversions with air quality monitoring, the development of new measurement types) could help overcoming the remaining challenges.

*Acknowledgements.* This study was conducted within the European CarboCountCity project funded by European Institute of Technology's Climate KIC program. Grégoire Broquet's research was funded and supported by the Chaire industrielle BridGES, a joint research program between Thalès Alenia Space, Veolia, and the parent institutions of LSCE (CEA, CNRS, UVSQ). Thanks are also due to Balendra Thiruchittampalan and Felix R. Vogel for providing the IER inventory. The latter we also thank for providing the temporal profiles to disaggregate this inventory's annual emissions to hourly emissions.

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

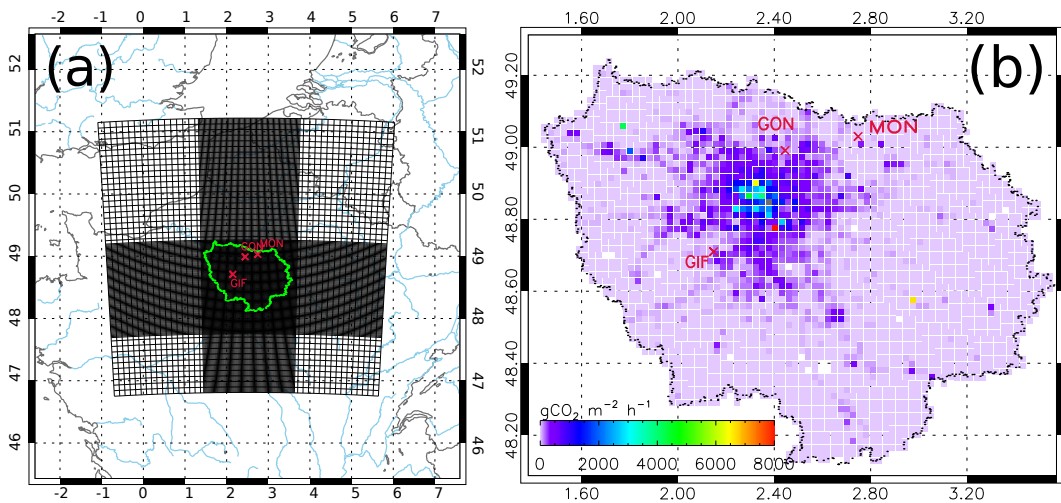

**Figure 1.** (a) Map of the Northern France modelling domain. The monitoring sites are depicted as red crosses, while the green line denotes the boundaries of the Île-de-France region. Black lines show the model grid: $2 \, \mathrm{km} \times 2 \, \mathrm{km}$ spatial resolution in the centre, $2 \, \mathrm{km} \times 10 \, \mathrm{km}$ and $10 \, \mathrm{km} \times 10 \, \mathrm{km}$ spatial resolution for the surroundings, respectively. (b) Fossil fuel $CO_2$ emission for January 2011 for each grid cell of the Île-de-France region according to AIRPARIF 2008.

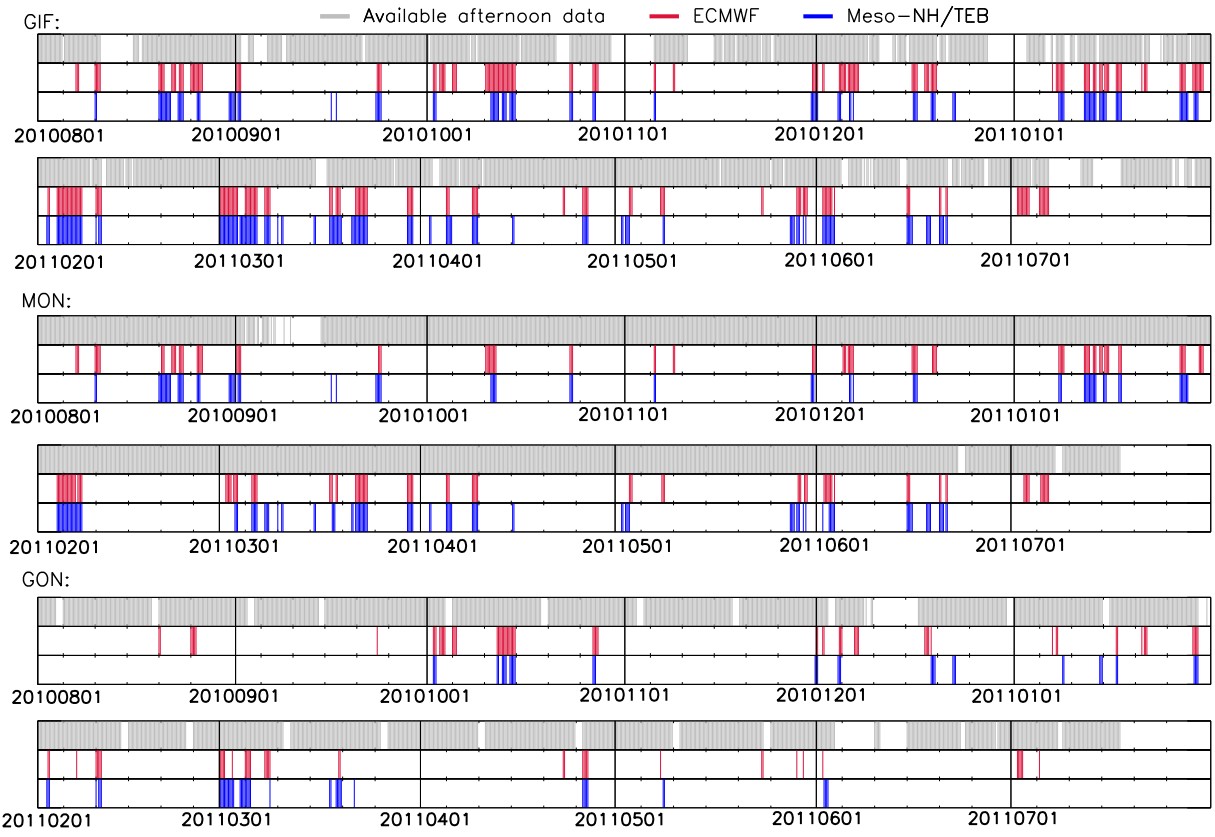

**Figure 2.** Afternoon (12-16 h) $CO_2$ data availability during the CO2-MEGAPARIS project (August 2010–July 2011) for the different monitoring sites used in this study. Available hourly observed data are displayed as grey vertical lines. Red and blue: observations that are actually assimilated when using the reference (stringent) gradient selection criteria. Red: Selection when using ECMWF wind fields for the wind estimation. Blue: Selection when using Meso-NH/TEB wind fields for the wind estimation.

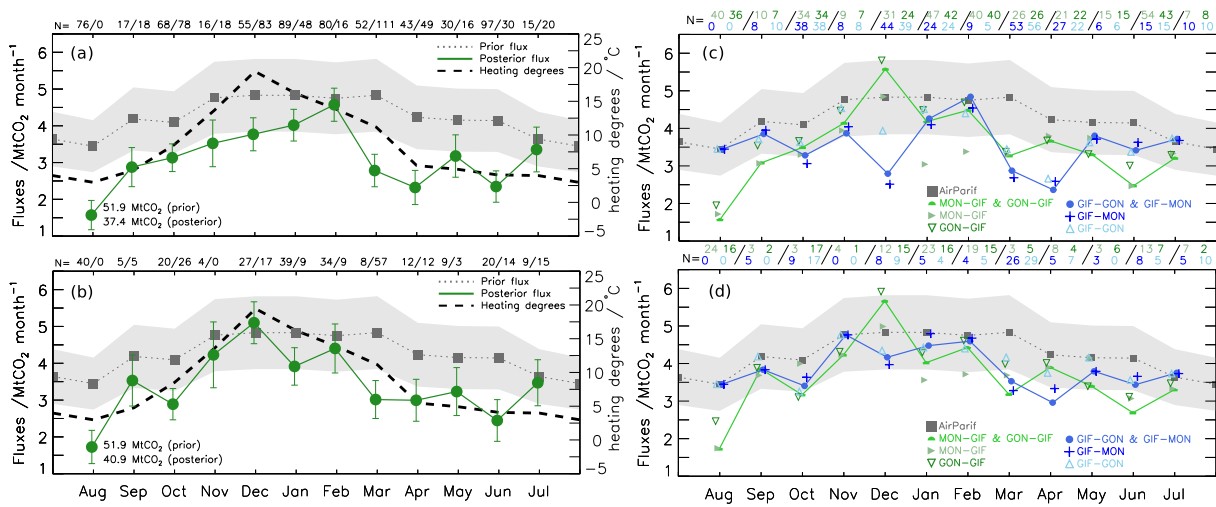

**Figure 3.** Monthly fossil fuel $CO_2$ emissions prior and posterior estimates from inversions in $MtCO_2$. Left column: Prior estimates (line) $\pm$ the standard deviation of uncertainties (shade) are displayed in grey while posterior estimates (line) $\pm$ the standard deviation (bars) are displayed in green. (**a**): Results using the initial inversion configuration. (**b**): Results using the reference inversion configuration. Monthly mean heating degrees for the centre of Paris, obtained from ECMWF's operational analysis, are displayed in black. Numbers at the top are those of the $CO_2$ mole fraction gradients assimilated for SW- or NE winds, respectively. Prior and posterior annual emission estimates are displayed in the left bottom corner. Right column: Results using the initial (**c**) and reference (**d**) configuration of the inversions but assimilate only subsets of selected gradients (see Sect. 2.2.) Colour-coded numbers at the top are those of the assimilated gradient by each subset. Prior estimates (line) $\pm$ the standard deviation of uncertainties (shade) are displayed in grey. Symbols are slightly shifted to prevent overlap.

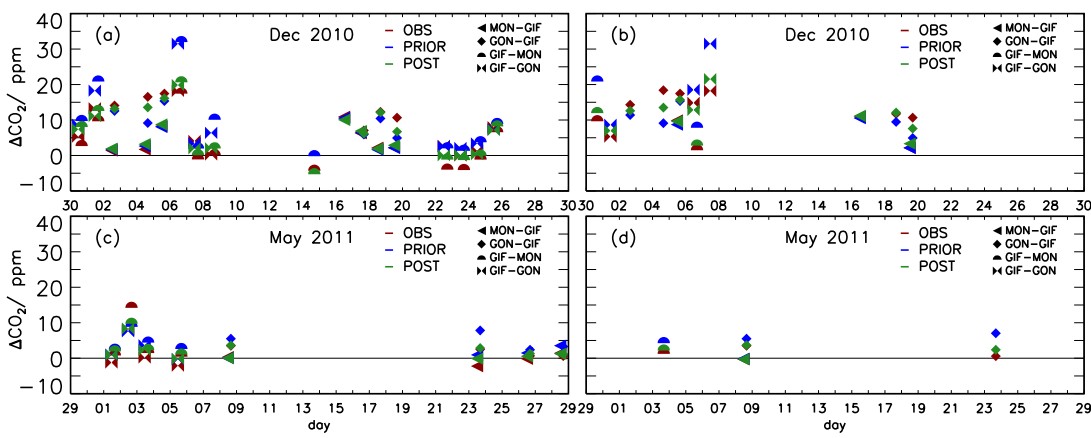

**Figure 4.** Time series of mean afternoon (12–16 h) $CO_2$ mole fraction differences ($\Delta CO_2$) between downwind and upwind sites for December 2010 (upper panel) and May 2011 (lower panel). Measured $CO_2$ mole fraction differences are shown in red; prior and posterior $CO_2$ mole fraction differences in blue and green, respectively. (**a**) and (**c**): Selection of gradients relies on wind direction ranges of the initial inversion configuration. (**b**) and (**d**): Wind direction ranges are limited to a narrow SW/NE wind corridor across the city following the reference inversion configuration (see Sect. 2.2).

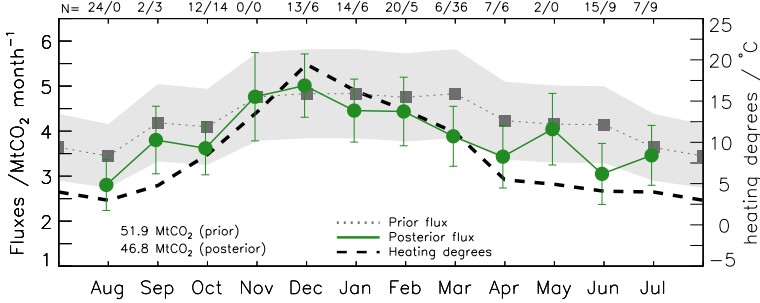

**Figure 5.** Results for the Lagrangian experiment (see Sect. 2.2 and Tab. 1). As for Fig. 3c, but a 2-hour time lag between downwind and upwind measurements is introduced.

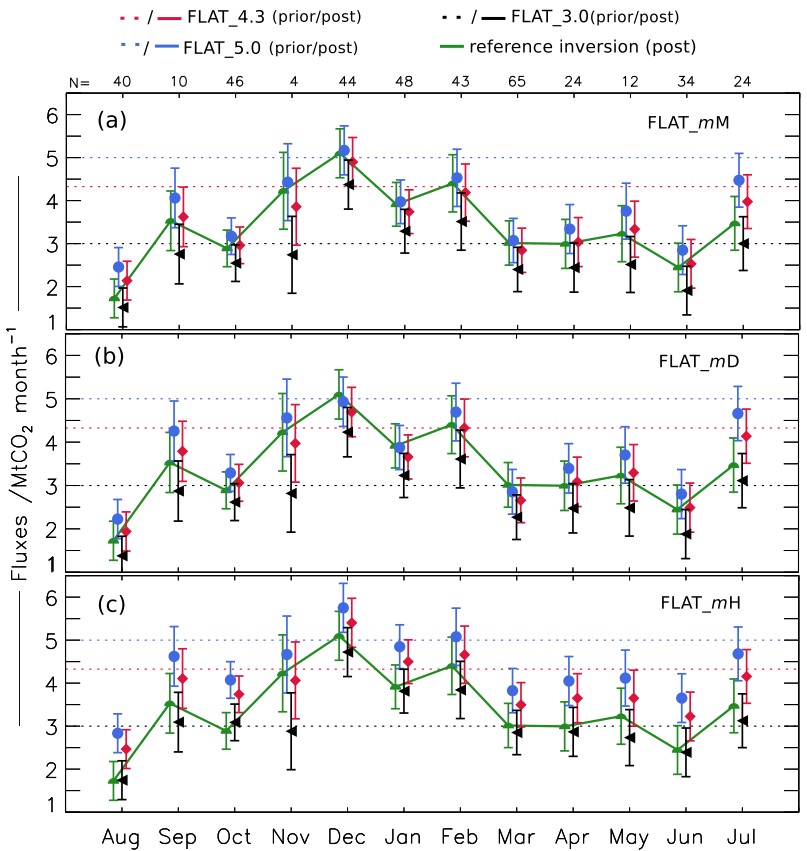

**Figure 6.** Sensitivity of monthly fossil fuel $CO_2$ emissions upon $\mathbf{x}_b$. Monthly fossil fuel $CO_2$ estimates $\pm$ the standard deviation of their uncertainties are shown for inversions that use $3\,\mathrm{MtCO_2 month^{-1}}$ (black), $4.3\,\mathrm{MtCO_2 month^{-1}}$ (red), and $5\,\mathrm{MtCO_2 month^{-1}}$ (blue) monthly prior emissions. **(a)** Priors are flat at monthly scale (FLAT_$m$M, $m$=3.0, 4.3 or 5.0 $\mathrm{MtCO_2 month^{-1}}$). **(b)** Priors are flat at daily scale (FLAT_$m$D). **(c)** Priors are flat at 6-h scale (FLAT_$m$H), see Sect. 2.6 for details). Fluxes obtained by the reference inversion are displayed in green. Numbers at the top denote the number of assimilated $CO_2$ mole fraction gradients. Symbols are slightly displaced to prevent overlap.

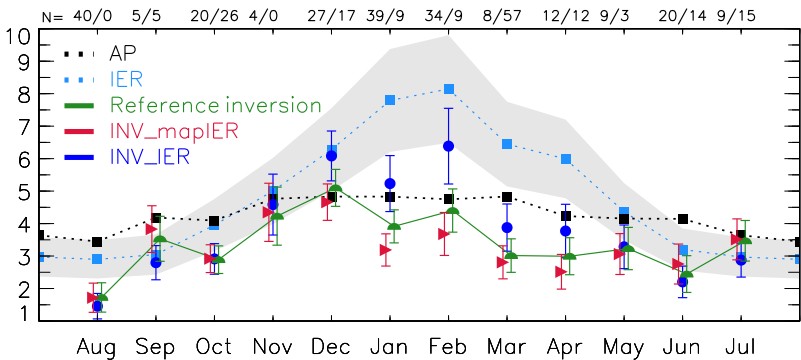

**Figure 7.** Sensitivity of monthly fossil fuel $CO_2$ emissions upon $\mathbf{H}^{\mathrm{map}}$. Red: Monthly fossil fuel $CO_2$ emissions estimates $\pm$ the standard deviation of their uncertainties obtained from the reference inversion (green), INV_mapIER (red), and INV_IER (blue), respectively. Monthly fossil fuel $CO_2$ emissions prior estimates by AIRPARIF are depicted in black while IER's monthly estimates $\pm$ the standard deviation of uncertainties are depicted in sky blue and grey, respectively. Note the different scale of the ordinate compared to Fig. 3, 5 and 6.

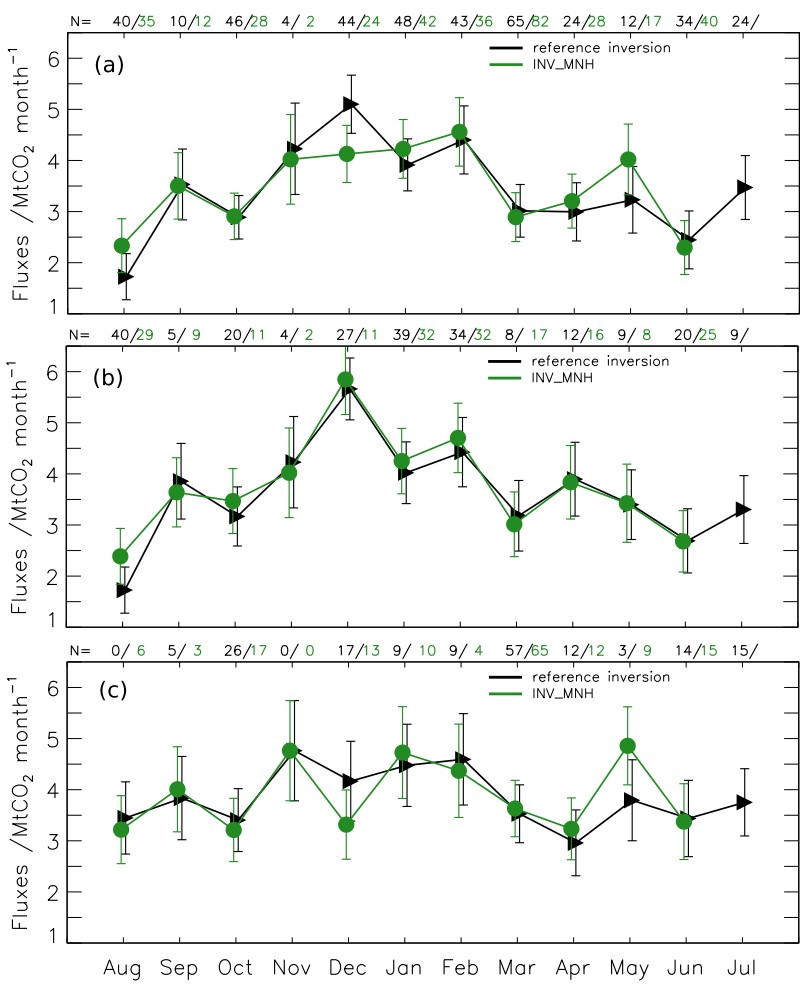

**Figure 8.** Sensitivity of monthly fossil fuel $CO_2$ budgets upon meteorological data. Displayed are the estimates $\pm$ the standard deviation of their uncertainties obtained from the reference inversion (green) and INV_MNH (black), respectively. Numbers at the top denote color-coded the number of assimilated gradients. **(a)** Assimilation of both SW and NE gradients. **(b)** Assimilation of SW gradients. **(c)** Assimilation of NE gradients.

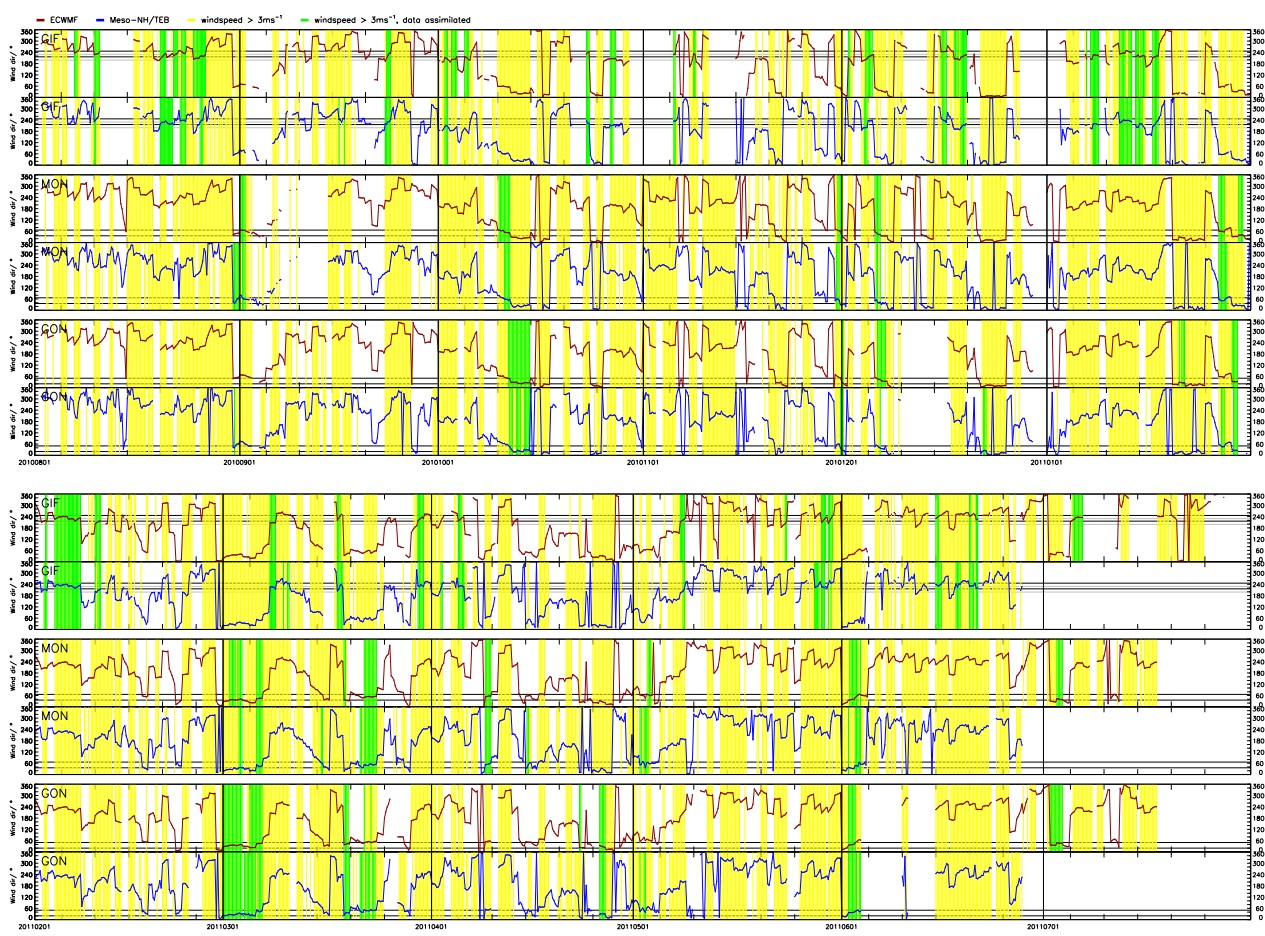

**Figure A1.** Time series of mean wind directions during afternoon (12-16 h) at the different monitoring sites used in this study. Solid horizontal lines denote the range of wind directions used by the reference (stringent) gradient selection (see Sect. 2.2). Red: Wind directions as simulated by ECMWF. Blue: Wind directions as simulated by Meso-NH/TEB. Yellow vertical lines indicate wind speed $>3\,\mathrm{ms}^{-1}$. Green vertical lines: Data are actually assimilated when using the reference (stringent) gradient selection criteria.

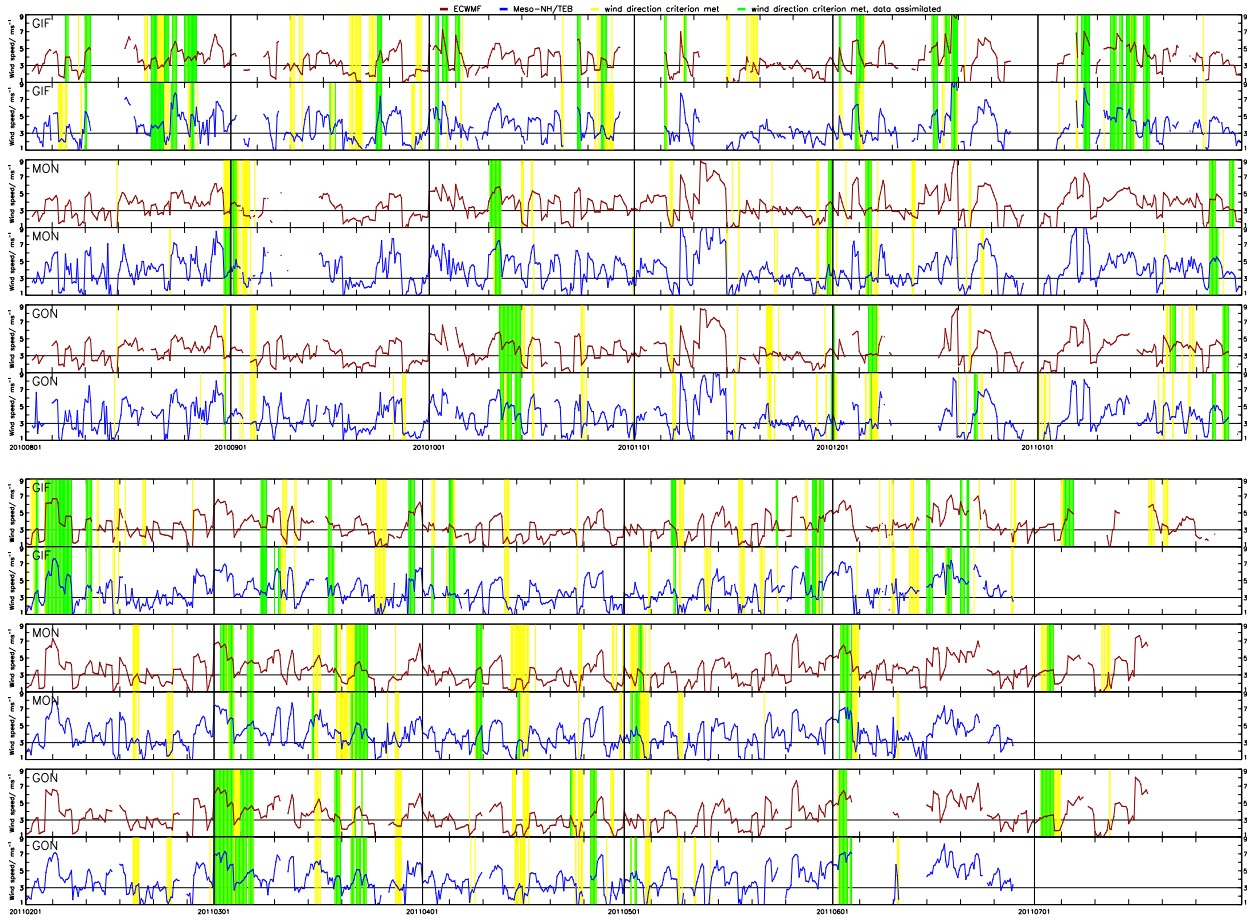

**Figure A2.** Time series of mean wind speed during afternoon (12-16 h) at the different monitoring sites used in this study. Solid horizontal lines denote a $3\,\mathrm{ms}^{-1}$ wind speed threshold. Red: Wind speed as simulated by ECMWF. Blue: Wind speed as simulated by Meso-NH/TEB. Yellow vertical lines indicate wind directions are within the range of wind direction used by the reference (stringent) gradient selection. Green vertical lines: Data are actually assimilated when using the reference (stringent) gradient selection criteria.

Table 1. Summary of the different inversion configuration and Île-de-France (IdF) annual fossil fuel $CO_2$ emissions from different inventories and inversion results. Priors that are flat at the monthly, daily and 6-hourly scale are denoted M, D and H, respectively (see section 2.6 for the details). Posterior estimates are derived from inversions using the operator and prior estimate indicated in the corresponding line of the table.

| Inversion | Acronym | H | | | $\mathbf{y}^{\mathrm{f}}$ | $\mathbf{x}^{\mathrm{b}}$ | IdF annual fossil fuel $CO_2$ emissions in $\mathrm{MtCO_2}$ | |
| --- | --- | --- | --- | --- | --- | --- | --- | --- |
| | | $\mathbf{H}^{\mathrm{samp}}$ | $\mathbf{H}^{\mathrm{trans}}$ | $\mathbf{H}^{\mathrm{map}}$ | | | Prior | Post |
| Initial | *ini* | $\mathbf{H}^{\mathrm{samp}}_{ini-ECM}$ | $\mathbf{H}^{\mathrm{trans}}_{ECM}$ | $\mathbf{H}^{\mathrm{map}}_{AP}$ | $\mathbf{y}^{\mathrm{f}}_{ini-ECM}$ | AP08 | 51.9 | 37.4 |
| Reference | *ref* | $\mathbf{H}^{\mathrm{samp}}_{ref-ECM}$ | $\mathbf{H}^{\mathrm{trans}}_{ECM}$ | $\mathbf{H}^{\mathrm{map}}_{AP}$ | $\mathbf{y}^{\mathrm{f}}_{ref-ECM}$ | AP08 | 51.9 | 40.9 |
| Sensitivity | FLAT_4.3H | | | | | H | 51.9 | 47.1 |
| Tests | FLAT_4.3D | | | | | D | 51.9 | 41.1 |
| | FLAT_4.3M | | | | | M | 51.9 | 41.4 |
| | FLAT_3.0H | $\mathbf{H}^{\mathrm{samp}}_{ref-ECM}$ | $\mathbf{H}^{\mathrm{trans}}_{ECM}$ | $\mathbf{H}^{\mathrm{map}}_{AP}$ | $\mathbf{y}^{\mathrm{f}}_{ref-ECM}$ | H | 36.0 | 37.1 |
| | FLAT_3.0D | | | | | D | 36.0 | 33.0 |
| | FLAT_3.0M | | | | | M | 36.0 | 33.0 |
| | FLAT_5.0H | | | | | H | 60.0 | 52.2 |
| | FLAT_5.0D | | | | | D | 60.0 | 45.3 |
| | FLAT_5.0M | | | | | M | 60.0 | 45.3 |
| | INV_mapIER | $\mathbf{H}^{\mathrm{samp}}_{ref-ECM}$ | $\mathbf{H}^{\mathrm{trans}}_{ECM}$ | $\mathbf{H}^{\mathrm{map}}_{IER}$ | $\mathbf{y}^{\mathrm{f}}_{ref-ECM}$ | AP08 | 51.9 | 39.0 |
| | INV_IER | $\mathbf{H}^{\mathrm{samp}}_{ref-ECM}$ | $\mathbf{H}^{\mathrm{trans}}_{ECM}$ | $\mathbf{H}^{\mathrm{map}}_{IER}$ | $\mathbf{y}^{\mathrm{f}}_{ref-ECM}$ | IER | 60.1 | 45.5 |
| | INV_MNH | $\mathbf{H}^{\mathrm{samp}}_{ref-MNH}$ | $\mathbf{H}^{\mathrm{trans}}_{MNH}$ | $\mathbf{H}^{\mathrm{map}}_{AP}$ | $\mathbf{y}^{\mathrm{f}}_{ref-MNH}$ | AP08 | 51.9 | [1] |
| Time lag | *lag* | $\mathbf{H}^{\mathrm{samp}}_{ref-ECM}$ | $\mathbf{H}^{\mathrm{trans}}_{ECM}$ | $\mathbf{H}^{\mathrm{map}}_{IER}$ | $\mathbf{y}^{\mathrm{f}}_{ref-ECM}$ | AP08 | 51.9 | 46.8 |
| Emissions for the year 2010 as given by AIRPARIF (2013) | | | | | | | | 41.8 |

[1] Meso-NH/TEB data are available up to June 2011 only

**Table 2.** Annual and seasonal bias, standard deviation (STD), root mean square error (RMSE) and coefficient of determination ($r^2$) of prior model-data misfit and posterior model-data misfit for the initial inversion (experiment $ini$) and the reference inversion (experiment $ref$), respectively. All values, except for $r^2$, are given in ppm.

| | Bias | | | | STD | | | | RMSE | | | | $r^2$ | | | |
| | $ini$ | | $ref$ | | $ini$ | | $ref$ | | $ini$ | | $ref$ | | $ini$ | | $ref$ | |
| | prior | post | prior | post | prior | post | prior | post | prior | post | prior | post | prior | post | prior | prior |
|---|---|---|---|---|---|---|---|---|---|---|---|---|---|---|---|---|
| Annual | 2.50 | 0.33 | 3.04 | 0.36 | 3.60 | 2.21 | 3.77 | 2.20 | 4.38 | 2.23 | 4.84 | 2.22 | 0.53 | 0.80 | 0.53 | 0.81 |
| JJA | 2.20 | 0.23 | 2.70 | 0.39 | 2.31 | 1.59 | 2.54 | 1.62 | 3.18 | 1.60 | 3.70 | 1.66 | 0.13 | 0.45 | 0.03 | 0.34 |
| SON | 2.41 | 0.28 | 3.73 | 0.38 | 3.49 | 1.98 | 2.95 | 2.05 | 4.23 | 2.00 | 4.74 | 2.07 | 0.35 | 0.75 | 0.27 | 0.61 |
| DJF | 2.35 | 0.48 | 2.79 | 0.44 | 4.21 | 2.51 | 4.55 | 2.29 | 4.82 | 2.55 | 5.33 | 2.32 | 0.61 | 0.84 | 0.55 | 0.85 |
| MAM | 3.01 | 0.26 | 3.29 | 0.20 | 3.65 | 2.38 | 4.01 | 2.63 | 4.72 | 2.39 | 5.17 | 2.62 | 0.22 | 0.56 | 0.07 | 0.50 |