# Peer review of "A first year-long estimate of the Paris region fossil fuel $CO_2$ emissions based on atmospheric inversion"

_Atmospheric Chemistry and Physics, 2016_

## Referee Comment (RC1) · J. Turnbull (Referee) · 3 May 2016

Review of A first year-long estimate of the Paris region fossil fuel CO2 emissions base don atmospheric inversion (Staufer et al)

This paper uses atmospheric CO2 observations in a Bayesian inversion to evaluate urban CO2 emissions for Paris. It builds on earlier work by Breon et al (2015) that first developed the inversion framework, using the innovative concept of inverting for differences between two observing sites (rather than absolute CO2 mole fractions). This work expands the dataset to a full year, which allows meaningful evaluation of how well the framework works. The inversion framework already shows some useful outcomes that can inform/improve the bottom-up inventory priors. That is, the observations imply

a stronger seasonal cycle in emissions than is represented in the bottom-up data products, and this stringer seasonal cycle is also consistent with expectations. They also discuss the challenges and limitations of their inversion framework. A major challenge is that the inversion result is still strongly dependent by the prior (bottom-up) emission estimate. They show that there is still much work to be done to provide detailed information from this type of regional inversion and they provide useful suggestions as to how the inversion could be improved.

This is a very nice paper and an excellent contribution to the (still very small) urban greenhouse gas literature. It is entirely appropriate for publication in ACP with some minor revisions as noted below.

General comments: There is very little attention paid to the contribution of the biosphere to the urban $CO_2$ observations and it's contribution to uncertainties in the inversion. The biosphere fluxes used as priors are described only very briefly, but there is no information about the quality of that prior and how much biases and uncertainties in it might contribute to biases/uncertainties in the inversion. Some discussion of this is needed in the paper.

Specific comments: Pg 2 line 14. "two-month" not "two-months"

Pg 2 line 26. Presumably air parcels pass over the city, rather than through it?

Pg 3. Lines 24-29. This paragraph is hard to follow. It transitions abruptly from a description of preliminary tests to describing what is in specific sections of the paper.

Pg 3 section 2. It would be helpful if the inversion parameters were referred to by what they are, rather than by the algebra term in the equations. For the reader who is not a specialist in Bayesian inversions, it is hard to remember what y yo, x etc are referring to.

Pg 5 lines 1-4. How good is C-TESSEL for the urban area? See also my general comment above.

Pg 5 lines 15-18. What is the measurement quality for the 1 hour means that are used in the inversion?

Pg 6 lines 3-4. There are other studies that show that ATMs do poorly at night. It would be better to reference some work other from outside your own research group.

Pg 6 lines 30-32. The justification for discarding the GON-MON gradients needs to be given. Is it that the sites are too close together so that emissions are not large enough to create consistent enhancements? Or is there a major flaw in the methodology?

Pg 6 line 26. "BR2015" should be "Breon et al 2015" I think.

Pg 7 plines 10-14. One of the gradients had far more impact than the rest, so was excluded? This seems important - what is the justification for excluding it? How different was the inversion when this gradient was included?

Pg7 lines 20-31. There are a number of minor typos in this section.

Pg 7 llines 20-31. Ylag experiment. How does the ylag experiment account for the evolving boundary layer during the day?

Pg 8 lines 16-21. This paragraph should come before rather than after the preceding one.

Pg 8 lines 29-31. Are these the emissions that are in the model domain but outside Ile-de-France? Not clear.

Pg 10 line 1. (first sentence). Is the conclusion you state from your work or elsewhere? If the latter, please reference.

Pg 11 lines 17-19. Please reference the independent analysis that shows the temperature dependence.

Pg 11 lines 27-32. This paragraph seems spurious.

Pg 12 lines 22-24. Can a problem with a bias in one observation site during the month

of December be ruled out as a cause of this December anomaly?

Pg 13 "r2" not "R2" in several places.

Pg 16 line 22. You have not demonstrated that GON-MON gradients are "not evidently related to the whole city emissions". See also my earlier comment.

Pg 16 lines 24-31. nd pg 17 lines 18-20. Have you tried inverting with the AIR-PARIF2010 prior? Does it pull the posterior values down even further? Or not? If every inversion pulls the values lower, does this imply a fundamental flaw in the inversion?

Pg 17 lines 1-2. "a large fraction of the Paris emissions are due to domestic and commercial heating". Add "believed to be" or something similar.

Pg 17 lines 29-31. This is a very awkward sentence.

Pg 18 lines 15-18. Be clear here that your inversion solves only for mid-pm observations, and your analysis does not exclude that the poor representation of the diurnal cycle could have a strong impact on nighttime emission estimates.

Pg 20 lines 13-15. Presumably actual nighttime measurements would be useful to constrain nighttime emissions as well!

Pg 20 line 18. The mention of Recife seems irrelevant.

Pg 20 line 25. What is GB?

Figure 3. Caption is inconsistent with labelling on graph (a, b, etc). Also, on the right hand panels, the numbers at the top are hard to follow (they are fine on the left panel).

---

## Referee Comment (RC2) · Anonymous Referee #2 · 6 Jul 2016

**1 Overview:**

Review of "*A first year-long estimate of the Paris region fossil fuel CO$_2$ emissions based on atmospheric inversion*" by Staufer et al.

Staufer et al. present a year-long estimate of CO$_2$ emissions in Paris using the "Gradient Method" developed by Breon et al., ACP (2015). This manuscript is largely just an extension of the Breon et al. (2015) paper. They now use a longer record of observations but they include fewer sites and use more stringent data selection. The manuscript is fairly well written. However, I have some serious concerns with the manuscript. In particular, the authors need to justify some of the crucial assumptions

in the "Gradient Method".

**2 Major comments:**

I have some major concerns with assumptions made in the "Gradient Method".

**2.1 Spatio-temporal offset between upwind and downwind measurements**

The authors assume, as in Breon et al. (2015), that the difference between measurements made at two of their surface sites are directly comparable and that the difference between them can be related to the city-scale emissions. However, it's not clear to me that this is a valid assumption. My issues with this assumption are: (1) a temporal lag between the measurements, (2) a spatial offset between the measurements, and (3) a poor choice of model to evaluate this. The critical question is: "Did the downwind measurement actually originate near the upwind site?"

*Temporal Lag:* The authors touch on this issue in the final paragraph of Section 3.2. The upwind and downwind sites are quite far apart in space and it will take a few hours for the airmass to travel from the upwind site to the downwind site. I did a quick back of the envelope calculation for the transport time using distances from Breon et al. (2015; Table 1). GIF is 23 km from the Paris centre, GON is 16 km from the Paris centre and the EIF site is directly between them. This would suggest these sites are about 40 km apart (Google Earth puts the distance at about 39km). A 3 m s$^{-1}$ windspeed (the minimum windspeed criteria) would give a transit time of about 4 hours, quite a bit longer than the temporal lag of 2 hours reported in Staufer et al. (although it's not clear how they estimate their temporal lag). This means that the upwind measurement should be compared with an observation that is at least 2-4 hours later in order to relate
it to the city-scale emissions (assuming there were no changes in wind direction over a 2-4 hour period).

However, a 4 hour difference in the observed airmass will make it difficult, if not impossible (without a model), to relate the two measurements because $CO_2$ has a strong diurnal cycle (see, for example, Fig. 1 from McKain et al., 2012 for the Salt Lake City diurnal cycle). It would be incorrect to compare measurements made at different hours because they are influence by external processes (like boundary layer growth).

The authors do touch on this issue in the last paragraph of Section 3.2 and show that it is a problem (inversion is not consistent with their other inversions) but disregard it anyway.

*Spatial Offset (vertical mixing):* It is unlikely that the airmass measured by the downwind instrument was actually in the direct vicinity of the upwind instrument, there is almost certainly some spatial offset. The author's data selection criteria are designed to minimize the horizontal offset between the downwind airmass and the upwind instrument but there is also probably a vertical offset. As mentioned above, the downwind airmass will take a few hours to travel from the upwind site to the downwind site and there was almost certainly some vertical motion during this period. It's possible that the downwind airmass was not even in the boundary layer when it passes the upwind instrument.

This leads into my next point.

*Poor choice of model to evaluate this:* The authors are using an Eulerian model, CHIMERE, for their atmospheric transport. However, a Lagrangian Particle Dispersion Model (LPDM), such as FLEXPART or WRF-STILT, would be more appropriate for this work and would allow them to answer the critical question from above ("Did the downwind measurement actually originate near the upwind site?"). An LPDM will

simulate trajectories for individual measurements which would allow the authors to easily determine if the downwind measurement actually originated near the upwind measurement. This would greatly simplify the "data selection criteria" because the authors would just need to find a minimum distance between the downwind trajectory and the upwind measurement site. It would also provide an appropriate time-lag for each measurement.

Instead, the authors have chosen to develop a set of "data selection criteria" that are extremely difficult to evaluate because they do not actually know the trajectories of their measurements (e.g., they spend a lot of the text arguing that the Breon et al. criteria are not stringent enough). It also means that they end up throwing out most (92%) of their data. . .

**2.2 Aggregation Error and Design of the Control Vector**

The authors use a very crude control vector. It is a single scaling factor for the entire Paris region (with some temporal resolution). This means that they assume the gradient between their two sites is representative of the ENTIRE Paris region and that the entire Paris region should be scaled up or down. As the authors mention (Page 6, Line 25), this will induce large aggregation errors because different parts of the city can no longer vary independently.

The authors acknowledge that this is a problem in Section 4.2 when they say: "The inversion results, however, are significantly affected by changes in the emission distribution. It reveals the need to rely on robust, high resolution emission maps such as those produced by local agencies like AIRPARIF." It seems more likely that this issue is due to aggregation error. For example, the authors could deduce a large underestimate in emissions if the downwind airmass happened to pass over some point source that is missing in the inventory. Because the authors only have a single scaling factor for the entire Paris region, the whole city would be scaled up and the emissions would now be

overestimated. So the question is, "Are these gradients actually representative of the ENTIRE Paris region?". This is why groups typically solve for fluxes at high-resolution.

**2.3 Boundary Layer Heights**

Accurate simulation of the boundary layer heights is crucial for modelling urban $CO_2$. It controls the size of the box over which the emissions are mixed (e.g., $CO_2$ concentrations are lower during the daytime even though emissions are peaking). However, there is no discussion of the boundary layer heights in the model. Are they reasonable?

**2.4 Resolution and representativeness of the measurements, prior, and meteorology**

The authors use a fairly coarse resolution model (CHIMERE at 2km with meteorology at 15km resolution) even though previous work (e.g., McDonald et al., 2014) has shown that 1km is necessary to resolve highways. From the abstract of McDonald et al. (2014): "High $CO_2$ emission fluxes over highways become apparent at grid resolutions of 1 km and finer." Why is 2km sufficient here?

Along a similar vein, the authors use surface observations from 4 to 7 meters above ground level. Are these measurements actually representative of a 2km $\times$ 2km region (which is assumed since their grid is 2km $\times$ 2km)?

**2.5 Separating fossil fuel and biosphere fluxes**

You don't have measurements that would distinguish between the two. It seems that most of this separation is from assumptions you've made, not data. Further, it seems that there could be compensating errors because that separation is unconstrained. As such, it seems like the title might be overstepping. "Fossil fuel" should probably be

removed from the title.

Also, what do the biosphere fluxes look like in the different inversion cases? Are they unchanged or could they be compensating for some of the changes you're seeing?

**3  Minor comments:**

**3.1  Dimensions of your matrices?**

What are the dimensions of the different matrices (e.g., are $\mathbf{H}^{map}$, $\mathbf{H}^{trans}$, and $\mathbf{H}^{samp}$ all the same dimension? How are they multiplied together?) Even just reporting them in terms of something like: "$\mathbf{H}$ is an $n_{\mathbf{y}} \times n_{\mathbf{x}}$ matrix" (can use other notation) would be helpful.

**3.2  Clarify how you are constructing the $\mathbf{H}$ matrix**

From Section 2.3.1, it seems that you're doing a brute-force construction of the $\mathbf{H}$ matrix (e.g., perturbing each element in your control vector in a separate simulation), is that correct?

**4  Specific comments:**

Page 1, Line 4: Can you reliably estimate 6-h emissions with just 4 hours of obs?

Page 4, Line 5: Are these errors unbiased if the PBL height were wrong? Wouldn't that induce a systematic bias? Was that tested for?

Page 4, Eq. 1: This form assumes that $n_{\mathbf{y}} > n_{\mathbf{x}}$, is that true? Given the small number of observations, I would have guessed the other form would be more computationally efficient.

Page 4, Lines 26-27: However, the diurnal cycle is probably largely unchanged because you're only using afternoon observations. I doubt there's much change to any other time periods.

Page 5, Lines 28-31: What about vertical gradients? These sites are all in the boundary layer so the airmasses might be fundamentally different...

Page 6, Lines 17-26: Why not solve at a finer spatial scale? This would greatly reduce the aggregation error.

Page 7, Lines 27-28: How do you estimate this? I get 3-4 hours using the cutoff windspeed of 3 m s$^{-1}$ (see Major Comment 2.1).

Page 7, Lines 30-31: Why not just do a more traditional inversion with finer spatial resolution? You could jointly solve for the background concentration (see, for example, Henne et al., 2016).

Page 9, Lines 25-31: Confusing. A plot of your covariance would be useful. Even just plotting a single row (would just be a simply x-y line plot) would be really helpful here.

Page 13, Lines 34-35: Misleading. It sounds like you're using a Lagrangian Particle

Dispersion Model, but I'm pretty sure you aren't since there was no mention of one.

Pages 14, Lines 1-4: This is what you should be doing though!

Pages 19, Lines 30-35: Again, I think this is the correct way to use the gradient method. I don't think a lack of data is a good justification for not using it. I think that you should either: (a) use the gradient method and the lag approach, (b) provide a better justification for why you don't need to consider temporal lags, or (c) use a traditional inversion with finer spatial resolution (with time-dependent footprints).

Pages 20, Lines 11-12: This is what most groups already do... A more traditional Bayesian inversion with an LPDM would allow you to solve at high spatial resolution without inducing large aggregation errors.

Figure 3: (b) and (c) labels should be flipped because they don't agree with the caption.

Measurement Sites: I don't think the paper lists the height of the measurement sites. I had to go back to Breon et al. (2015) to find it.

**5   References:**

Henne *et al.*: Validation of the Swiss methane emission inventory by atmospheric observations and inverse modelling, *Atmos. Chem. Phys.*, doi:10.5194/acp-16-3683-2016, 2016.

McKain *et al.*: Assessment of ground-based atmospheric observations for verification of greenhouse gas emissions from an urban region, *Proc. Nat. Acad. Sci.*, doi:10.1073/pnas.1116645109, 2012.

McDonald *et al.*: High- resolution mapping of motor vehicle carbon dioxide emissions, *J. Geophys. Res. Atmos.*, doi:10.1002/2013JD021219, 2014.

---

## Author Comment (AC1) · 21 Sep 2016

**Reply to comments by Jocelyn Turnbull**

Review of A first year-long estimate of the Paris region fossil fuel CO2 emissions based on atmospheric inversion (Staufer et al)

This paper uses atmospheric CO2 observations in a Bayesian inversion to evaluate urban CO2 emissions for Paris. It builds on earlier work by Breon et al (2015) that first developed the inversion framework, using the innovative concept of inverting for differences between two observing sites (rather than absolute CO2 mole fractions). This work expands the dataset to a full year, which allows meaningful evaluation of how well the framework works. The inversion framework already shows some useful outcomes that can inform/improve the bottom-up inventory priors. That is, the observations imply a stronger seasonal cycle in emissions than is represented in the bottom-up data products, and this stringer seasonal cycle is also consistent with expectations. They also discuss the challenges and limitations of their inversion framework. A major challenge is that the inversion result is still strongly dependent by the prior (bottom-up) emission estimate. They show that there is still much work to be done to provide detailed information from this type of regional inversion and they provide useful suggestions as to how the inversion could be improved.

This is a very nice paper and an excellent contribution to the (still very small) urban greenhouse gas literature. It is entirely appropriate for publication in ACP with some minor revisions as noted below.

We thank Dr. J. Turnbull for this positive assessment of our study, for her comments, and for having highlighted our objective of discussing the challenges, limitations and potential improvement of the current framework as well as the first successes obtained with it. We consider that it is a critical aspect of our analysis (see answers to reviewer 2).

General comments: There is very little attention paid to the contribution of the biosphere to the urban CO2 observations and it's contribution to uncertainties in the inversion. The biosphere fluxes used as priors are described only very briefly, but there is no information about the quality of that prior and how much biases and uncertainties in it might contribute to biases/uncertainties in the inversion. Some discussion of this is needed in the paper.

We agree that the topic of the natural fluxes is critical for city scale inversions and we agree that we need to discuss it in this paper. However:

- Paris is a particular case in the sense that it is a very dense urban area with limited vegetation inside it.
- According to the model simulations, the computation of gradients succeeds in cancelling the impact of biogenic fluxes in the data that are analyzed (it was too rapidly mentioned in the last section of the paper).
- The weak impact of biogenic fluxes on the inversion of anthropogenic CO2 (according to the modelling framework) has already been addressed in more detail in Bréon et al. (2015), with which we try to avoid redundancies. In addition, the specific topic of the

cancelling of the biogenic component in the simulated CO2 gradients has also been analyzed in the Boon et al. 2015, paper.
-   The new computation of the gradients proposed in this study further decreases the potential impact of the biogenic fluxes in the inversion.

Therefore, we expand the introduction by discussing the cancelling of the natural component in the CO2 gradients according to the model. We will indicate in the result section that the amplitude of the natural component of the simulated CO2 gradients is further decreased when narrowing the wind ranges for the gradient selection. Furthermore, we will expand the discussion on this topic based on such an analysis and carefully remind that actual gradients in the measurements may bear a larger signature of natural fluxes than the simulated gradients. See also the answer to the specific comment on C-TESSEL.

Reference:

Boon, A., Broquet, G., Clifford, D. J., Chevallier, F., Butterfield, D. M., Pison, I., Ramonet, M., Paris, J.-D., and Ciais, P.: Analysis of the potential of near-ground measurements of CO2 and CH4 in London, UK, for the monitoring of city-scale emissions using an atmospheric transport model, Atmos. Chem. Phys., 16, 6735-6756, doi:10.5194/acp-16-6735-2016, 2016.

Specific comments: Pg 2 line 14. "two-month" not "two-months"

It will be corrected.

Pg 2 line 26. Presumably air parcels pass over the city, rather than through it?

Yes. It will be corrected in the revised manuscript.

Pg 3. Lines 24-29. This paragraph is hard to follow. It transitions abruptly from a description of preliminary tests to describing what is in specific sections of the paper.

We will add 1 or 2 sentences explaining that this revision consists in narrowing the wind range for the gradient selection to avoid situations during which the air leaving the "upwind site" or reaching the "downwind site" does not overpass a significant part of the city and the vicinity of the other site.

Pg 3 section 2. It would be helpful if the inversion parameters were referred to by what they are, rather than by the algebra term in the equations. For the reader who is not a specialist in Bayesian inversions, it is hard to remember what y yo, x etc are referring to.

We will add the name of the variable before such labels throughout section 2 wherever it does not alter the reading of the text.

Pg 5 lines 1-4. How good is C-TESSEL for the urban area? See also my general comment above.

This model is definitely not perfect for modeling ecosystems within and in the vicinity of urban

areas. It has a relatively low resolution compared to that of the transport model. This should, in principle, increase the challenge of dealing with the impact of uncertainties in the natural fluxes since significant NEE is simulated on grid cells with high emissions at the edges of the urban areas. However, the fact that the signature on the simulated gradients from such fluxes is low demonstrates that these drawbacks from the C-TESSEL product do not have critical impacts on the inversion. In the revised manuscript, this will be discussed based on references to Bréon et al. 2015 and Boon et al. 2015 when we present the C-TESSEL product.

Pg 5 lines 15-18. What is the measurement quality for the 1 hour means that are used in the inversion?

Based on the regular analysis of target gases, and intercomparison of side by side measurements (at Gif sur Yvette) the accuracy of the hourly averages at the three sites is better then 0.4 ppm which makes the measurement errors for the hourly averages negligible compared to the signals and other sources of errors discussed in this study.

Pg 6 lines 3-4. There are other studies that show that ATMs do poorly at night. It would be better to reference some work other from outside your own research group.

We will reference to Geels et al., ACP, 2007 in the revised manuscript.

References:

Geels, C., Gloor, M., Ciais, P., Bousquet, P., Peylin, P., Vermeulen, A. T., Dargaville, R., Aalto, T., Brandt, J., Christensen, J. H., Frohn, L. M., Haszpra, L., Karstens, U., Rödenbeck, C., Ramonet, M., Carboni, G., and Santaguida, R.: Comparing atmospheric transport models for future regional inversions over Europe – Part 1: mapping the atmospheric CO2 signals, Atmos. Chem. Phys., 7, 3461-3479, doi:10.5194/acp-7-3461-2007, 2007.

Pg 6 lines 30-32. The justification for discarding the GON-MON gradients needs to be given. Is it that the sites are too close together so that emissions are not large enough to create consistent enhancements? Or is there a major flaw in the methodology?

The explanation was given at Pg 6 lines 11-12: we definitely consider that the sites are too close from each other so that in the GON-MON gradients, we face uncertainties in the high resolution emission mapping rather than uncertainties in the city-cale budget of the emissions. Furthermore, the Roissy Charles de Gaulle airport and its very specific type of emissions (with very specific problems for simulating their injection heights) is located between the two sites. The text at this place will be slightly expanded in the revised manuscript.

Pg 6 line 26. "BR2015" should be "Breon et al 2015" I think.

Yes, it will be corrected.

Pg 7 plines 10-14. One of the gradients had far more impact than the rest, so was excluded? This seems important - what is the justification for excluding it? How different was the inversion when this gradient was included?

We provide a detailed answer to this technical topic below. However, as explained at the end of this answer, it has a negligible impact on the study.

We based our judgment on an objective computation of the "observation impact" which consists in indicating how much a given data, or more precisely a model-data misfit, drives the increment applied to the fluxes given a set of data assimilated (it is given by the product between the gain matrix K and $y0_i - H_i x_b$ where $y_i$ is the corresponding observation). Through the least square minimization of the misfits to the data, it can happen that one data point out of tens of data points is the dominating driver of the inversion results. It happens, for instance, if the corresponding model-data misfit is far larger than the other ones, or if at the corresponding time, the atmospheric transport is such that the sensitivity of this data point to the fluxes is far larger than that of the other data points (typically if the PBL is very shallow at the corresponding time). The large model-data misfit can reveal a very high observation error and giving too much weight to a single data is dangerous (we prefer to work with situations where there is a more balanced fit to most of the data). For similar reasons, inverse modelers generally filter out data corresponding to model data misfits that are more than several times the standard deviation of the whole set of model-data misfits they use (e.g., Chevallier et al 2010). Looking at extreme observation impacts can be viewed as an alternative way to detect situations where we could give an excessive weight to large observation errors.

We made such an analysis when running a preliminary *ini* inversion. We actually removed two data points in November (instead of 1 data point in November and 1 data point in December as erroneously said in the manuscript). The data points removed had both an impact of nearly -0.3MtCO2 on the emission budget for November (i.e. ~ -0.6MtCO2 in total). The impact of other data during this month were not more negative than -0.1MtCO2 (such a situation was extreme as demonstrated by the fact that we removed 2 data points only). In both cases, this was driven by high prior model-data misfits in combination with relatively low vertical mixing conditions in November.

Opposed to what the manuscript could have let think, these data correspond to wind directions for which the gradients are not selected in the reference configurations of the observation vector. Therefore, the impact of this removal applies to the *ini* experiment only, and it is quite negligible for our study.

We will expand a bit the text to discuss this but given the weak impact it has for the results, we will keep it short.

Reference:

Chevallier, F., P. Ciais, T. J. Conway, T. Aalto, B. E. Anderson, P. Bousquet, E. G. Brunke, L. Ciattaglia, Y. Esaki, M. Fröhlich, A.J. Gomez, A.J. Gomez-Pelaez, L. Haszpra, P. Krummel, R. Langenfelds, M. Leuenberger, T. Machida, F. Maignan, H. Matsueda, J. A. Morguí, H. Mukai, T. Nakazawa, P. Peylin, M. Ramonet, L. Rivier, Y. Sawa, M. Schmidt, P. Steele, S. A. Vay, A. T. Vermeulen, S. Wofsy, D. Worthy, 2010: CO2 surface fluxes at grid point scale estimated from a global 21-year reanalysis of atmospheric measurements. J. Geophys. Res., 115, D21307,- doi:10.1029/2010JD01388

Pg7 lines 20-31. There are a number of minor typos in this section.

We will correct them.

Pg 7 llines 20-31. Ylag experiment. How does the ylag experiment account for the evolving boundary layer during the day?

The boundary layer (BL) does not evolve only in time, but also in space. The concern about a varying BL for the gradients was raised in a more general way during the review of the study of Bréon et al., 2015 (see the discussion with reviewer 1 http://www.atmos-chem-phys.net/15/1707/2015/acp-15-1707-2015-discussion.html). Because of the variability of the boundary layer, the gradients by themselves cannot be a perfect representation of the enrichment of $CO_2$ in the air when crossing the city, even though we expect it to be a good proxy for it. But, in any case, the assimilation of gradients better characterizes it than the assimilation of individual $CO_2$ measurements for any wind direction, especially since the signature of remote fluxes on the atmospheric $CO_2$ concentrations is expected to be well mixed and does not to evolve much with BL variations over short durations and distances.

Above all, our atmospheric transport model simulates this variability of the boundary layer and therefore the inversion accounts for it when assimilating the gradients. The uncertainty in the modelling of the boundary layer is part of the model errors. Such a model error is a traditional source of error in inverse modelling systems, which we do not overcome by assimilating gradients. The role of setting-up the R matrix in the inverse modelling system is to account for such errors.

This topic will now be discussed earlier when we recall the concept of gradients in the introduction but we will mainly refer to the similar discussions in Bréon et al. 2015.

Pg 8 lines 16-21. This paragraph should come before rather than after the preceding one.

We will reverse the order of the paragraphs.

Pg 8 lines 29-31. Are these the emissions that are in the model domain but outside Ile-de-France? Not clear.

It is a combination of the influence of these emissions, and that of the model boundary and initial conditions (any component of the model that is not controlled by the inversion). The beginning of the sentence at line 31 and the phrase "emissions outside Île de France" instead of "emissions outside Île-de-France but within the modeling domain" are indeed misleading.

We will clarify these sentences in the revised manuscript.

Pg 10 line 1. (first sentence). Is the conclusion you state from your work or elsewhere? If the latter, please reference.

It comes from our own work. We will clarify it in the revised manuscript.

Pg 11 lines 17-19. Please reference the independent analysis that shows the temperature dependence.

We will clarify the fact that this analysis has been conducted by one of the co-authors of this paper (François-Marie Bréon) so that we cannot cite it as a personal communication.

To complement our answer, here are some details about the analysis he produced and which has not been published. The figure below shows the variations of the electric consumption in the Île-de-France region as a function of the temperature in 2013 at 15:00. Similar figures for the other hours of the day demonstrate a similar threshold of the temperature at ~19° below which the consumption is highly (negatively) correlated to the temperature (with similar slopes for each hour). The electricity consumed in Île-de-France is mainly generated by nuclear power plants. Given their clear dependence on atmospheric temperature, these variations, however, very likely reveal variations in heating behavior. And, a large part of heating in Île-de-France is based on gas consumption, which is responsible for a large part of the $CO_2$ emissions in the Paris area. Assuming that users of gas and electric heaters display similar heating behavior, this result supports the assumption of the temperature dependence described in the manuscript.

[Figure]

Pg 11 lines 27-32. This paragraph seems spurious.

We will shorten it. We will remind how the uncertainties shown in the figures are derived and will not warn about the fact the uncertainties will not be analyzed in detail (this will be explained in the discussion only). A sentence will be added on the uncertainties arising from the inversions at the end of section 3.2.

Pg 12 lines 22-24. Can a problem with a bias in one observation site during the month of December be ruled out as a cause of this December anomaly?

We do not think so since both types of gradients GIF-MON and GIF-GON drive the inversion in the same way (see the next sentence at l.25). We thus cannot connect it to a bias at MON or GON. And a local bias at a given site would impact SW gradients too and would probably impact the site during other months.

Pg 13 "r2" not "R2" in several places.

We will apply this correction.

Pg 16 line 22. You have not demonstrated that GON-MON gradients are "not evidently related to the whole city emissions". See also my earlier comment.

We think the small distance between these two sites and the fact that emissions between two sites are significantly impacted by the Roissy Charles-de-Gaulles airport is sufficient to assume that the gradients between these two sites cannot be representative of the city-scale emissions.

We will remind in this sentence that the improvement is assessed through the analysis of the inverted emissions.

Pg 16 lines 24-31. nd pg 17 lines 18-20. Have you tried inverting with the AIRPARIF2010 prior? Does it pull the posterior values down even further? Or not? If every inversion pulls the values lower, does this imply a fundamental flaw in the inversion?

We did not have access to the spatialized inventory corresponding to the AIRPARIF (2013) report on the emissions for 2010 (in which we extracted the so-called AIRPARIF2010 annual data) for this study. Still, we have conducted tests with prior values lower than that of AIRPARIF2008 in the FLAT_3.0 experiments and it is one of the topic of the paragraph on p17 line 18 to p18 line 2.

In the case of FLAT_3.0H, the inversion increases the emissions, so that our system does not systematically pulls the values lower. Our argument regarding the decrease of the annual emissions in the FLAT_3.0M experiment is that this decrease, in practice, does not arise from a systematic decrease of the monthly emissions, and that it is quite small (so that we have a strong convergence from the ensemble of prior values to the ensemble of posterior values obtained in the set of experiments).

In the paragraph p17 line18 to p18 line2 we will more clearly separate the discussions on a.) the potential impact of using relative prior uncertainties in the convergence of the results, and b.) the decrease of the annual emissions when using the day-to-day variations from AIRPARIF2008.

Pg 17 lines 1-2. "a large fraction of the Paris emissions are due to domestic and commercial heating". Add "believed to be" or something similar.

We will change the parenthesis (43%) into (43% according to the AIRPARIF2008 inventory) in this sentence.

Pg 17 lines 29-31. This is a very awkward sentence.

We will reformulate it.

Pg 18 lines 15-18. Be clear here that your inversion solves only for mid-pm observations, and

your analysis does not exclude that the poor representation of the diurnal cycle could have a strong impact on nighttime emission estimates.

Actually, we must acknowledge an error when running the FLAT_mD experiments. The actual results from FLAT_mD are far closer to those from FLAT_mM than to those from FLAT_mH, opposed to what was indicated in the manuscript. This is far more logical given that the inversion can control 6-hour mean fluxes and thus, to some extent, the day to day variations. As highlighted by the reviewer, it, however, cannot control nighttime emissions except through the indirect extrapolation of the information driven by the correlations in the prior error covariance. Still, of note is that the hourly mean emissions for the 11h-16h window is only ~30% above the hourly mean emissions for the whole day in the AIRPARIF2008 inventory.

We will correct the results of experiment FLAT_mD and make a clear conclusion regarding the impact of the description of diurnal cycle in the prior estimate of the 6-hour mean budgets.

Pg 20 lines 13-15. Presumably actual nighttime measurements would be useful to constrain nighttime emissions as well!

Initially, the sentence assumed that, in principle, it should be easier to model parts of the PBL's transitional phases in late morning and late afternoon, during which we have some level of mixing, than to model the PBL at nighttime. But, such a discussion would be too loose at this point and we agree with this comment by the reviewer. The sentence will be rewritten to encompass nighttime data.

Pg 20 line 18. The mention of Recife seems irrelevant.

We will remove it.

Pg 20 line 25. What is GB?

This refers to Gregoire Broquet.

Figure 3. Caption is inconsistent with labelling on graph (a, b, etc). Also, on the right hand panels, the numbers at the top are hard to follow (they are fine on the left panel).

Thanks for pointing to the inconsistent figure labels. It will be corrected.

---

## Author Comment (AC2) · 21 Sep 2016

**Reply to comments by Anonymous Reviewer #2:**

We thank reviewer 2 for her/his technical review of our manuscript. It gives us the opportunity to improve the clarity of our text, since most of the major comments were already discussed in the submitted version of the manuscript and seem to come from a misunderstanding of the main concept underlying our assimilation of gradients. We hope that the following additional explanations will clarify the concept and results of our study.

**1 Overview:**

Review of "*A first year-long estimate of the Paris region fossil fuel $CO_2$ emissions based on atmospheric inversion*" by Staufer et al.

Staufer et al. present a year-long estimate of $CO_2$ emissions in Paris using the "Gradient Method" developed by Breon et al., ACP (2015). This manuscript is largely just an extension of the Breon et al. (2015) paper.

Our manuscript represents much more than "just an extension" of the Bréon et al. (2015) paper. As its title implied ("An attempt at…"), Bréon et al. (2015) showed first results of Paris-area $CO_2$ emission estimation from atmospheric measurements during a few weeks, based on a gradient approach. The present study brings deeper analysis of the concept of assimilating cross-city gradients and evaluates the inversions strengths and weaknesses with the results from a full year worth of measurements, an improved method, and a series of sensitivity tests to the main components of the inverse modeling system. Technically, it represented a large amount of work (for instance, we had to rebuild the **H** matrix in the inverse modeling system with the Meso-NH/TEB-CHIMERE transport configuration). We also paid a lot of attention not to be redundant with Bréon et al. (2015) in terms of analysis and discussions in the manuscript. The two papers should rather be read as two complementary studies.

They now use a longer record of observations but they include fewer sites and use more stringent data selection. The manuscript is fairly well written. However, I have some serious concerns with the manuscript. In particular, the authors need to justify some of the crucial assumptions in the "Gradient Method".

As detailed by the answers below, in principle, there is no reason to think that our gradient method is more prone to errors than a more straight-forward inversion approach that would avoid adapting the observation and control vectors to the weaknesses of the models. Both this paper and Bréon et al. (2015) actually produce a series of analyses to verify that the gradient approach definitely helps to overcome some practical limitations of the traditional inverse modelling framework.

This will be better stated in the discussion section of the revised manuscript.

**2 Major comments:**

I have some major concerns with assumptions made in the "Gradient Method".

2.1 Spatio-temporal offset between upwind and downwind measurements

The authors assume, as in Breon et al. (2015), that the difference between measurements made at two of their surface sites are directly comparable and that the difference between them can be related to the city-scale emissions. However, it's not clear to me that this is a valid assumption. My issues with this assumption are: (1) a temporal lag between the measurements, (2) a spatial offset between the measurements, and (3) a poor choice of model to evaluate this.

These issues were already discussed in the paper. (1) was addressed by the introduction of a time-lag between upwind and downwind measurements that did not improve the results (Figure 5, Section 3.2). (2) was addressed by the use of a new strict wind selection (Section 3.2). (3) was addressed by changing the meteorological product (from ECMWF to Meso-NH/TEB) for the forcing of CHIMERE and the gradient selection (Section 3.3.3).

The critical question is: "Did the downwind measurement actually originate near the upwind site?"

This is not a critical question for our concept of relating the cross-city gradients to the city-scale emissions. As explained in the introduction and section 2.2 of the manuscript, this concept assumes that we cancel or at least decrease the impact of fluxes outside the city by assimilating cross-city gradients instead of individual measurements (because we assume that this impact has a relatively large spatial and temporal scale), and that the information about the city emissions, that is contained in the assimilated cross-city gradients, corresponds to a large part of the city rather than to just small portions of it.

Ensuring that the downwind measurement correspond to an air mass that actually originates near the upwind site at the time of the upwind measurement would help to verify the first assumption (which is why we have conducted the test case "*lag*" with a time-lag between the upwind and downwind data in the gradients). However, this is not a requirement since:

- the impact of the boundary conditions, if not also that of a major part of fluxes outside the city, can be assumed to be quite smooth in time and thus quite similar over a 2-h window (see the debate about this specific time-lag below) due to atmospheric mixing

 - we do not need a perfect canceling of the impact of boundary conditions when assimilating gradients instead of individual measurements to improve the inversion behavior.

Both the Bréon et al. 2015 paper and our study demonstrate that even though our approach is not perfect (in the sense that there likely remains some impact of the remote fluxes in the measured gradients) it yields better results than if assimilating individual measurements for any wind direction.

This discussion will be expanded in the new manuscript (in the introduction, method and

discussion sections).

This reply and that to the general comment 1 answer in a general way the following questions within comment 2.1. Our answers to these following questions focus on the specific technical points that they raise, without systematically reminding what is said above.

*Temporal Lag:* The authors touch on this issue in the final paragraph of Section 3.2.

Actually the discussion on this topic starts in the last paragraph of Section 2.2 and ends at the beginning of Section 4.3.

The upwind and downwind sites are quite far apart in space and it will take a few hours for the airmass to travel from the upwind site to the downwind site. I did a quick back of the envelope calculation for the transport time using distances from Breon et al. (2015; Table 1). GIF is 23 km from the Paris centre, GON is 16 km from the Paris centre and the EIF site is directly between them. This would suggest these sites are about 40 km apart (Google Earth puts the distance at about 39km). A 3 m s$^{-1}$ windspeed (the minimum windspeed criteria) would give a transit time of about 4 hours, quite a bit longer than the temporal lag of 2 hours reported in Staufer et al. (although it's not clear how they estimate their temporal lag).

3 m/s is the minimum wind speed near the surface (i.e. at station height) for the assimilated gradients. Over the year, the average wind speed near the surface is about 4.5 m/s. Looking at the wind speed near the surface is relevant when aiming at avoiding too much influence of sources near the measurement sites. The transport through the city, however, should be better characterized by the average wind speed in the PBL. We actually selected a 2-hour time-lag because the average wind speed 100m above the ground is about 7 m/s over the year.

It will be clarified in the revised manuscript.

This means that the upwind measurement should be compared with an observation that is at least 2-4 hours later in order to relate it to the city-scale emissions (assuming there were no changes in wind direction over a 2-4 hour period).

However, a 4 hour difference in the observed airmass will make it difficult, if not impossible (without a model), to relate the two measurements because $CO_2$ has a strong diurnal cycle (see, for example, Fig. 1 from McKain et al., 2012 for the Salt Lake City diurnal cycle).

Here, we use an inventory with diurnal temporal profiles and an atmospheric transport model to relate the measurements and to account for such a diurnal cycle. These models are not perfect. However, uncertainties in the modeled diurnal cycle of the emissions and in the atmospheric transport (which relates emissions at a location and time and concentrations at another location and a later time, i.e. the basic concept of atmospheric inversion) impact any of the existing inverse modelling approaches. There is no reason to assume that it would be a more critical issue for the gradient simulation.

It would be incorrect to compare measurements made at different hours because they are

influence by external processes (like boundary layer growth).

The reviewer use the term "compare" in a misleading way. The difference between the two measurements is interpreted with the help of a transport model in the same way that emissions at a given time are connected to measurements at later times and at other locations with the help of a transport model in the basic concept of the inversion.

See also the answer to the first reviewer who focuses on the boundary layer:

"The boundary layer does not evolve only in time, but also in space. The concern about a varying BL for the gradients was raised in a more general way during the review of the study by Bréon et al. (2015) (see the discussion with reviewer 1 http://www.atmos-chem-phys.net/15/1707/2015/acp-15-1707-2015-discussion.html). Because of the variability of the boundary layer, the gradients by themselves cannot be a perfect representation of the enrichment of CO2 in the air when crossing the city even though we expect it to be a good proxy for it. But, in any case, the assimilation of gradients better characterizes it than the assimilation of individual CO2 measurements for any wind direction, especially since the signature of remote fluxes on the atmospheric CO2 concentrations is expected to be well mixed and not to evolve much with BL variations over short durations and distances.

Above all, our atmospheric transport model simulates this variability of the boundary layer and therefore the inversion accounts for it when assimilating the gradients. The uncertainty in the modelling of the boundary layer is part of the model errors. Such a model error is a traditional source of error in inverse modelling systems, which we do not overcome by assimilating gradients. The role of setting-up the R matrix in the inverse modelling system is to account for such errors.

In the revised manuscript, this topic will be discussed earlier when we recall the concept of gradients in the introduction. We, however, mainly refer to similar discussions by Bréon et al. 2015."

The authors do touch on this issue in the last paragraph of Section 3.2 and show that it is a problem (inversion is not consistent with their other inversions) but disregard it anyway.

We think that our results (and analysis in the corresponding text) show that it is not a problem of inconsistency, but rather a problem of a weak constraint from the reduced datasets when a time-lag in the gradients is considered. Qualitatively, the corrections are consistent between experiments *ref* and *lag* but the amplitude of the corrections is smaller in the latter. In the revised manuscript, the text will be slightly modified to explain this in a clearer way.

*Spatial Offset (vertical mixing):* It is unlikely that the airmass measured by the downwind instrument was actually in the direct vicinity of the upwind instrument, there is almost certainly some spatial offset.

Such a (random) offset should not be an issue as long as the variations in time and space of the impact of fluxes outside the Paris area on CO2 concentrations are relatively small over periods and distance that correspond to the typical "space and time offsets" that are tolerated by the

method. If these variations were large, the gradient computation would only fail to decrease this impact compared to  using a traditional observation vector.

The author's data selection criteria are designed to minimize the horizontal offset between the downwind airmass and the upwind instrument

Indeed, the choice of a significant range of wind directions for the gradient selection acknowledges the acceptance for such an offset, even if the amplitude of the horizontal atmospheric diffusion should be considered in such a debate. This offset is not only taken into account by the study, but its amplitude is also discussed in the manuscript through discussions on the range of wind directions to be used for the gradient selection.

but there is also probably a vertical offset. As mentioned above, the downwind airmass will take a few hours to travel from the upwind site to the downwind site and there was almost certainly some vertical motion during this period. It's possible that the downwind airmass was not even in the boundary layer when it passes the upwind instrument.

See all previous answers above and in particular the one concerning the variations in the boundary layer.

This leads into my next point.

*Poor choice of model to evaluate this:*

The authors are using an Eulerian model, CHIMERE, for their atmospheric transport. However, a Lagrangian Particle Dispersion Model (LPDM), such as FLEXPART or WRF-STILT, would be more appropriate for this work and would allow them to answer the critical question from above ("Did the downwind measurement actually originate near the upwind site?").

Eulerian and Lagrangian models are two types of atmospheric transport models, with both having pros and cons. To our knowledge, there has not been any study demonstrating that one is "more appropriate" than the other one at the spatial scale and resolution of our study. Note that the Eulerian CHIMERE model has been selected to produce the official operational forecasts of air pollution in the Paris area and a large number of studies on the chemistry-transport of pollutants in the Paris region have been based on this model.

Technically, Lagrangian models are also not exclusively required for computing so-called emission and concentration "footprints". The adjoint of the CHIMERE Eulerian model (e.g. Broquet et al., 2011) could be used for the computation of the atmospheric "footprints" of the downwind measurements.

Whatever tool is used, answering the reviewer's question ("Did the downwind measurement actually originate near the upwind site?") is not straightforward because of the atmospheric diffusion and because of model uncertainties. There is not a clear threshold of the sensitivity of the downwind measurements to the concentration at the upwind site above (below) which we should say that the corresponding air mass travelled (did not travel, respectively) from the upwind to the downwind site. And, the uncertainties in the state-of-the-art meteorological

forcing is such (with typically more than 10° errors on the hourly wind direction at a given location; Lac et al. 2013, discussion on Bréon et al. 2015) that it would be vain to expect, based on footprint computations, a precise metric of whether "the downwind measurement actually originate near the upwind site". (Accounting for such modeling uncertainties motivated some of the sensitivity studies that we have presented and the analysis of the selected gradients when using different models as meterological forcing).

An LPDM will simulate trajectories for individual measurements which would allow the authors to easily determine if the downwind measurement actually originated near the upwind measurement.

The reviewer assumes that LPDM provides perfect simulations of the actual transport and we disagree on this point (see above).

This would greatly simplify the "data selection criteria" because the authors would just need to find a minimum distance between the downwind trajectory and the upwind measurement site. It would also provide an appropriate time-lag for each measurement.

By speaking about a "minimum distance" (with a "distance" which would likely be difficult to define precisely), the reviewer acknowledges the fact that it would be quite impossible to gather a significant number of gradients for which "the downwind measurement actually originates near the upwind site". This further decreases the legitimacy of the reviewer's assumption that the use of adjoint transport simulations would be far better than wind analysis to check this. The minimum distance proposed by the reviewer corresponds to the "offsets" that she/he criticizes earlier.

Instead, the authors have chosen to develop a set of "data selection criteria" that are extremely difficult to evaluate because they do not actually know the trajectories of their measurements (e.g., they spend a lot of the text arguing that the Breon et al. criteria are not stringent enough).

On the contrary, the data selection criteria can be evaluated in terms of the cancelling of the components of the natural fluxes and boundary conditions in the simulated $CO_2$ gradients, and through the evaluation of the inversion results. The discussions on the fact that the Breon et al. criteria is not stringent enough rely on the simple fact that this one does not ensure that air masses reaching the downwind site have crossed a major part of the city at all.

 It also means that they end up throwing out most (92%) of their data...

The filtering would be even larger if we would be willing to ensure that "the downwind measurement actually originate near the upwind site" whatever  transport model is used to evaluate it. And, if "offsets" and "minimum distances" are tolerated, the rate of selected data would be correlated to the corresponding level of tolerance.

2.2 Aggregation Error and Design of the Control Vector

The authors use a very crude control vector. It is a single scaling factor for the entire Paris region (with some temporal resolution). This means that they assume the gradient between their

two sites is representative of the ENTIRE Paris region and that the entire Paris region should be scaled up or down. As the authors mention (Page 6, Line 25), this will induce large aggregation errors because different parts of the city can no longer vary independently.

The authors acknowledge that this is a problem

Rather than naming it "problem" we prefer to remind that it is a source of uncertainty (such as transport errors) which is accounted for in both the set-up of the R matrix (see section 2.5) and when conducting a test of sensitivity to the spatial distribution of the emissions.

in Section 4.2 when they say: "The inversion results, however, are significantly affected by changes in the emission distribution. It reveals the need to rely on robust, high resolution emission maps such as those produced by local agencies like AIRPARIF." It seems more likely that this issue is due to aggregation error.

The two points are strongly connected. By looking at the mismatch when using two different maps of the emissions, we actually characterize the aggregation error. While the sensitivity to these maps is significant (as highlighted by the citation of our discussion section by the reviewer), the differences between the results from *ref* and INV_mapIER is not dramatic enough to form the basis for a major concern regarding our control vector.

For example, the authors could deduce a large underestimate in emissions if the downwind airmass happened to pass over some point source that is missing in the inventory. Because the authors only have a single scaling factor for the entire Paris region, the whole city would be scaled up and the emissions would now be overestimated.

Unlike what is sometimes assumed, solving for the emissions at the grid scale does not prevent from bearing aggregation errors. The use of relatively long spatial (and temporal) correlations in the prior uncertainty covariance matrix for the uncertainty in grid-scale emissions results in errors that correspond to aggregation errors. Such a use of long correlations, however, is necessary to regularize the inversion and to extrapolate the information from very scarce observation networks.

In the example given by the reviewer, given our observation network, a grid-scale inversion would not detect and solve for the missing point source. It would scale up the emission in an area corresponding to the typical spread of the measurement emissions footprints inflated by the prior error correlation length scale.

Our choice of the control vector implies an aggregation over the whole Ile de France region. But, almost all of the emissions in this region concentrate within the ~40 km x 40 km Paris urban area.

More importantly, our choice of the control vector is in line with our selection of the observation vector (which is an important aspect of our study): by assimilating "cross-city gradients" we minimize the impact of aggregation errors (see below) while we still account for them in the diagnostic or the **R** observation error covariance matrix from Bréon et al., who use gradients that were representative of smaller parts of the city. From this point of view, the

example given by the reviewer does not fit: the characteristics of the emissions in the Paris area (dominated by widespread sources) and the assimilation of cross-city gradients prevent the mis-location of a given point source from impacting the inversion in the way described by the reviewer.

We will slightly expand our discussions on the aggregation errors in the revised manuscript by adding part of the above answer in the introduction and discussion sections.

So the question is, "Are these gradients actually representative of the ENTIRE Paris region?".

The question is rather, "are these gradients actually representative of the Paris urban area ?" (see above). It is difficult to give an objective answer to this question. Still, the comparison of the results from experiments *ref* and INV_mapIER demonstrate that we can answer in a rather positive way.

This is why groups typically solve for fluxes at high-resolution.

See above for the answer regarding groups solving fluxes at high resolution but use long spatial correlation scales for the prior uncertainty in the emissions. We assume that the reviewers refer to inverse modeling groups in general since there have been very few attempts at conducting city-scale inversions. To our knowledge,  there has been only one attempt at solving for the emissions at "high resolution" (i.e. Lauvaux et al., JGR, 2016).

From our experience, the city scale inversion activity requires adapting the traditional framework of the inversion at larger scales. As discussed in the final section of the paper, we propose some strategies to overcome the main issues that we first encountered, and  showed first promising results. We hope that future observation frameworks and modeling techniques will allow us for solving the emissions at a higher resolution and for assimilating much more data. But we assume that it will require time, resources and research.

References:

Lauvaux, T., et al. (2016), High-resolution atmospheric inversion of urban CO2emissions during the dormant season of the Indianapolis Flux Experiment (INFLUX), J. Geophys. Res. Atmos., 121, 5213–5236, doi:10.1002/2015JD024473.

2.3 Boundary Layer Heights (PBLH)

Accurate simulation of the boundary layer heights is crucial for modelling urban $CO_2$. It controls the size of the box over which the emissions are mixed (e.g., $CO_2$ concentrations are lower during the daytime even though emissions are peaking). However, there is no discussion of the boundary layer heights in the model. Are they reasonable?

Lac et al. (2013) gave a positive assessment of the PBLH of their Meso-NH simulations (which

are used in our study) based on comparisons to PBLH measurements in the Paris area. Given the lack of PBLH observations, we did not resume such an analysis, neither in the Bréon et al. (2015) paper nor in this study. However, through our comparison of the results between using ECMWF and Meso-NH as forcing of CHIMERE, we provide an assessment of the impact of uncertainties in the meteorological simulation, which include uncertainties in the modeling of the PBLH (see Section 3.3.3). Finally, the diagnostic of the observation error by Bréon et al. (2015), which is used to set-up our inversion system, should also encompass uncertainties in the PBLH.

This will be better discussed in the revised manuscript.

2.4 Resolution and representativeness of the measurements, prior, and meteorology

The authors use a fairly coarse resolution model (CHIMERE at 2km

The use of the adjective "coarse" is highly subjective in this comment. In principle, Eulerian meteorological or transport models like WRF and CHIMERE can hardly be used at the subkm scale resolution. The type of models used for solving the meteorology-transport at higher resolution face the difficulty of simulating turbulent patterns which are quite impossible to fit.

with meteorology at 15km resolution)

In the INV_MNH experiment, we actually use a meteorological product at 2km resolution.

 even though previous work (e.g., McDonald et al., 2014) has shown that 1km is necessary to resolve highways. From the abstract of McDonald et al. (2014): "High $CO_2$ emission fluxes over highways become apparent at grid resolutions of 1 km and finer." Why is 2km sufficient here?

Over distances, the representation errors vanish due to atmospheric mixing. We do not need to resolve precisely the local impact of individual highways or chimneys (or individual cars and buildings) on concentrations when we target the emissions at the city-scale using peri-urban sites at the edges of the urban area.

Along a similar vein, the authors use surface observations from 4 to 7 meters above ground level. Are these measurements actually representative of a 2km × 2km region (which is assumed since their grid is 2km × 2km)?

These measurements are made in 2km × 2km "regions" at the edge of the urban area without any major CO2 source, and under wind conditions where the CO2 signal should be dominated by the signature of upwind fluxes (so that the representation error should be limited). Campaigns of dense arrays of measurements at high frequency across these 2km×2km "regions" would be needed to verify such an assumption. However, we do not have such campaigns to support this assumption. Still, the Desroziers diagnostic of the observation error used by Bréon et al. (2015) encompasses the potential representation error.

2.5 Separating fossil fuel and biosphere fluxes

You don't have measurements that would distinguish between the two. It seems that most of this separation is from assumptions you've made, not data. Further, it seems that there could be compensating errors because that separation is unconstrained. As such, it seems like the title might be overstepping. "Fossil fuel" should probably be removed from the title.

See the answer to a similar comment by the first reviewer:

"We agree that the topic of the natural fluxes is critical for city scale inversions and we agree that we need to discuss it in this paper. However:

- Paris is a particular case in the sense that it is a very dense urban area with limited vegetation inside it.
- According to the model simulations, the computation of gradients succeeds in cancelling the impact of biogenic fluxes in the data that are analyzed (it was too rapidly mentioned in the last section of the paper).
- The weak impact of biogenic fluxes on the inversion of anthropogenic CO2 (according to the modelling framework) has already been addressed in more detail in Bréon et al. (2015). In addition, the specific topic of the cancelling of the biogenic component in the simulated CO2 gradients has also been analyzed in the Boon et al. 2015, paper.
- The new computation of the gradients proposed in this study further decreases the potential impact of the biogenic fluxes in the inversion.

Therefore, we expand the introduction by discussing the cancelling of the natural component in the CO2 gradients according to the model. We will indicate in the result section that the amplitude of the natural component of the simulated CO2 gradients is further decreased when narrowing the wind range for the gradient selection. Furthermore, we will expand the discussion on this topic based on such an analysis and carefully remind that actual gradients in the measurements may bear a larger signature of natural fluxes than the simulated gradients."

Also, what do the biosphere fluxes look like in the different inversion cases? Are they unchanged or could they be compensating for some of the changes you're seeing?

By assimilating cross-city gradients, we impose a very weak constraint on the natural fluxes (which are really weak within the Paris urban area, see above). This topic was already analyzed by Bréon et al. 2015 and this is emphasized here by the use of narrower wind ranges for the gradient selection. This explains why we prefer not to analyze it in this manuscript. We will clarify it in the introduction of the revised manuscript.

**3 Minor comments:**

3.1 Dimensions of your matrices?

What are the dimensions of the different matrices (e.g., are $H^{map}$, $H^{trans}$, and $H^{samp}$ all the same dimension? How are they multiplied together?) Even just reporting them in terms of

something like: "H is an $n_y \times n_x$ matrix" (can use other notation) would be helpful.

The dimension of the different matrices will be provided in the revised manuscript.

3.2 Clarify how you are constructing the H matrix

From Section 2.3.1, it seems that you're doing a brute-force construction of the H matrix (e.g., perturbing each element in your control vector in a separate simulation), is that correct?

We compute the "response function" to each control variable. In the revised manuscript, we will detail this computation in a separate subsection of 2.3.

**4 Specific comments:**

Page 1, Line 4: Can you reliably estimate 6-h emissions with just 4 hours of obs?

This question is the main topic of our paper and of the Bréon et al. paper. In the absence of a more specific question, we can only refer the reviewer to these two papers.

Page 4, Line 5: Are these errors unbiased if the PBL height were wrong?

Yes, if the errors in PBL do not bias the simulation of CO2 gradients (typically if the modeled PBLH is unbiased).

Wouldn't that induce a systematic bias? Was that tested for?

We did not conduct sensitivity tests of the simulated CO2 as a function of different scenarios for errors in the PBL (if this is what the reviewer refers to). But we think that this is out of the scope of this study. See the answers to the generic questions above regarding errors in the simulated PBLH.

Page 4, Eq. 1: This form assumes that $n_y > n_x$, is that true?

Not necessarily. It just assumes that nx is small enough. Of note is that in the *ini* experiment, there are months during which ny>nx. The order of magnitude of ny and nx is the same in this study, so this is really a detail (see below).

Given the small number of observations, I would have guessed the other form would be more computationally efficient.

Since nx is small, it is extremely efficient as it is. We test different configuration of y in this study and as explained in the discussion section, there were plans to increase the network at the time of this study (and we have actually run inversions with more sites since the end of this study). This is why we preferred using the formulation of Eq.1 to write the code for the inversion system.

Page 4, Lines 26-27: However, the diurnal cycle is probably largely unchanged because you're

only using afternoon observations. I doubt there's much change to any other time periods.

Two of the 6-hour windows are directly constrained by the observations. There are some indirect constraints from the correlations in the B matrix. This was analyzed in the Bréon et al. 2015 paper and we will remind their conclusion regarding it.

Page 5, Lines 28-31: What about vertical gradients? These sites are all in the boundary layer so the airmasses might be fundamentally different. . .

The impact of remote fluxes diffuses horizontally and vertically upwind of the city. There is no reason to think that they will generate strongly different vertical gradients between the PBL and the free troposphere upwind vs downwind the city.

Page 6, Lines 17-26: Why not solve at a finer spatial scale? This would greatly reduce the aggregation error.

See the answer to the general question above about this topic.

Page 7, Lines 27-28: How do you estimate this? I get 3-4 hours using the cutoff windspeed of 3 m s$^{-1}$ (see Major Comment 2.1).

See the answer to this major comment above.

Page 7, Lines 30-31: Why not just do a more traditional inversion with finer spatial resolution? You could jointly solve for the background concentration (see, for example, Henne et al., 2016).

See the answer to the general comment above about this topic. In addition to this general answer: we preferred not to rush into controlling emissions at fine spatial resolution and parameters associated to the "background concentration" and turn a blind eye to the information content in the observations. Simple inverse modelling frameworks as the one proposed here help understanding and evaluating the issues and challenges associated with an inversion problem. City-scale inversion definitely bears a large number of challenges that we do not claim to solve at once (see our discussion). The blind faith in piling up control variables faces challenging practical issues if we do not run inversion systems as black boxes: how to define spatial correlations for uncertainties in anthropogenic emissions at 2km resolution without using abusive assumptions? How to reduce the number of control variables for the boundary conditions to a "manageable" number that could be constrained with the few sites available; this would easily raise large aggregation errors on these boundaries, whose impact can have large variations from day to day and over the distances of the boundaries. From our understanding, the method proposed by Henne et al. (2016) for controlling the boundary conditions as a smooth baseline would hardly cope with the day-to-day variations of the impact from the boundary conditions shown in Bréon et al (2015).

Page 9, Lines 25-31: Confusing. A plot of your covariance would be useful. Even just plotting a single row (would just be a simply x-y line plot) would be really helpful here.

The Kronecker product between the day-to-day and 6-hour window to 6-hour window would be impossible to understand in such a plot. We will rather rewrite the paragraph for clarification.

Page 13, Lines 34-35: Misleading. It sounds like you're using a Lagrangian Particle Dispersion Model, but I'm pretty sure you aren't since there was no mention of one.

Lagrangian properties do not belong to Lagrangian transport models only. But we agree that other readers could be confused by the term "Lagrangian inversion" and we will modify it in the revised manuscript.

Pages 14, Lines 1-4: This is what you should be doing though!

We assume that this comment is equivalent to the generic comment 2.1. Please see the answers to this comment.

Pages 19, Lines 30-35: Again, I think this is the correct way to use the gradient method. I don't think a lack of data is a good justification for not using it. I think that you should either: (a) use the gradient method and the lag approach, (b) provide a better justification for why you don't need to consider temporal lags, or (c) use a traditional inversion with finer spatial resolution (with time-dependent footprints).

Please see the answers to comment 2.1.

Pages 20, Lines 11-12: This is what most groups already do. . .

The sentence means that once we will be able to exploit urban measurements, we can plan to solve for–in the sense of targeting the estimation of–the spatial distribution of the emissions. As long as relying on peri-urban sites, we consider that we cannot rely on estimates that would be derived for sub-part of the city. In the revised manuscript, it will be rewritten for clarity.

A more traditional Bayesian inversion with an LPDM would allow you to solve at high spatial resolution without inducing large aggregation errors.

This crude assumption propagates the idea that atmospheric inversion is an objective technique which does not need to be precisely adapted to the specific challenges of a real case study, and that running after higher resolution always solves for the problems. We do not support this vision. Atmospheric inversion is a tool whose core parameters are informed by the user, and whose theoretical framework and technical limitations can make it extremely weak if the observation and control vectors are not well adapted.

Figure 3: (b) and (c) labels should be flipped because they don't agree with the caption.

Thank you for pointing to the inconsistent figure labels. It will be corrected.

Measurement Sites: I don't think the paper lists the height of the measurement sites. I had to go back to Breon et al. (2015) to find it.

We will provide these heights in the revised manuscript.